# On Learning Domain-Invariant Representations for Transfer Learning with Multiple Sources

**Trung Phung**[1] **Trung Le**[2] **Long Vuong**[1] **Toan Tran**[1] **Anh Tran**[1] **Hung Bui**[1] **Dinh Phung**[1,2]
[1] VinAI Research, Vietnam   [2] Monash University, Australia
`v.trungpq3@vinai.io, trunglm@monash.edu,`
`{v.longvt8, v.toantm3, v.anhtt152, v.hungbh1, v.dinhpq2}@vinai.io`

## Abstract

Domain adaptation (DA) benefits from the rigorous theoretical works that study its insightful characteristics and various aspects, e.g., learning domain-invariant representations and its trade-off. However, it seems not the case for the multiple source DA and domain generalization (DG) settings which are remarkably more complicated and sophisticated due to the involvement of multiple source domains and potential unavailability of target domain during training. In this paper, we develop novel upper-bounds for the target general loss which appeal to us to define two kinds of domain-invariant representations. We further study the pros and cons as well as the trade-offs of enforcing learning each domain-invariant representation. Finally, we conduct experiments to inspect the trade-off of these representations for offering practical hints regarding how to use them in practice and explore other interesting properties of our developed theory.

## 1   Introduction

Although annotated data has been shown to be really precious and valuable to deep learning (DL), in many real-world applications, annotating a sufficient amount of data for training qualified DL models is prohibitively labour-expensive, time-consuming, and error-prone. Transfer learning is a vital solution for the lack of labeled data. Additionally, in many situations, we are only able to collect limited number of annotated data from multiple source domains. Therefore, it is desirable to train a qualified DL model primarily based on multiple source domains and possibly together with a target domain. Depending on the availability of the target domain during training, we encounter either *multiple source domain adaptation* (MSDA) or *domain generalization* (DG) problem.

Domain adaptation (DA) is a specific case of MSDA when we need to transfer from a single source domain to an another target domain available during training. For the DA setting, the pioneering work [3] and other following work [43, 52, 53, 21, 18, 54, 22] are abundant to study its insightful characterizations and aspects, notably what *domain-invariant (**DI**) representations* are, how to learn this kind of representations, and the trade-off of enforcing learning DI representations. Those well-grounded studies lay the foundation for developing impressive practical DA methods [14, 47, 44, 52, 7, 51, 36].

Due to the appearance of multiple source domains and possibly unavailability of target domain, establishing theoretical foundation and characterizing *DI representations* for the MSDA and especially DG settings are significantly more challenging. A number of state-of-the-art works in MSDA and DG [56, 39, 19, 11, 41, 27, 28, 34, 15, 55, 49, 35, 37] have implicitly exploited and used DI representations in some sense to achieve impressive performances, e.g., [28, 28, 11] minimize representation divergence during training, [41] decomposes representation to obtain invariant feature, [55] maximizes domain prediction entropy to learn invariance. Despite the successes, the notion DI representations in MSDA and DG is not fully-understood, and rigorous studies to theoretically

35th Conference on Neural Information Processing Systems (NeurIPS 2021).

characterize DI representations for these settings are still very crucial and imminent. In this paper, we provide theoretical answers to the following questions: (1) what are DI representations in MSDA and DG, and (2) what one should expect when learning them. Overall, our contributions in this work can be summarized as:

- In Section 2.2, we first develop two upper-bounds for the general target loss in the MSDA and DG settings, whose proofs can be found in Appendix A. We then base on these bounds to characterize and define two kinds of DI representations: *general DI representations* and *compressed DI representations*.

- We further develop theory to inspect the pros and cons of two kinds of DI representations in Section 2.3, aiming to shed light on how to use those representations in practice. Particularly, two types of DI representations optimize different types of divergence on feature space, hence serving different purposes. Appendix B contains proofs regarding these characteristics.

- Finally, we study in Section 2.4 the trade-off of two kinds of DI representations which theoretically answers the question whether and how the target performance is hurt when enforcing learning each kind of representation. We refer reader to Appendix C for its proof.

- We conduct experiments to investigate the trade-off of two kinds of representations for giving practical hints regarding how to use those representations in practice as well as exploring other interesting properties of our developed theory.

It is worth noting that although MSDA has been investigated in some works [18, 54], our proposed method is the first work that rigorously formulates and provides theoretical analysis for representation learning in MSDA and DG. Specifically, our bounds developed in Theorems 1 and 3 interweaving both input and latent spaces are novel and benefit theoretical analyses in deep learning. Our theory is developed in a general setting of multi-class classification with a sufficiently general loss, which can be viewed as a non-trivial generalization of existing works in DA [30, 3, 53, 31, 54], each of which considers binary classification with specific loss functions. Moreover, our theoretical bounds developed in Theorem 8 is the first theoretical analysis of the trade-off of learning DI representations in MSDA and DG. Particularly, by considering the MSDA setting with single source and target domains, we achieve the same trade-off nature discovered in [53], but again our setting is more general than binary classification setting in that work.

## 2 Our Main Theory

### 2.1 Notations

Let $\mathcal{X}$ be a data space on which we endow a data distribution $\mathbb{P}$ with a corresponding density function $p(x)$. We consider the multi-class classification problem with the label set $\mathcal{Y} = [C]$, where $C$ is the number of classes and $[C] := \{1, \ldots, C\}$. Denote $\mathcal{Y}_\Delta := \left\{ \alpha \in \mathbb{R}^C : \|\alpha\|_1 = 1 \wedge \alpha \geq \mathbf{0} \right\}$ as the $C-$simplex label space, let $f : \mathcal{X} \mapsto \mathcal{Y}_\Delta$ be a probabilistic labeling function returning a $C$-tuple $f(x) = [f(x, i)]_{i=1}^C$, whose element $f(x, i) = p(y = i \mid x)$ is the probability to assign a data sample $x \sim \mathbb{P}$ to the class $i$ (i.e., $i \in \{1, ..., C\}$). Moreover, a domain is denoted compactly as pair of data distribution and labeling function $\mathbb{D} := (\mathbb{P}, f)$. We note that given a data sample $x \sim \mathbb{P}$, its categorical label $y \in \mathcal{Y}$ is sampled as $y \sim Cat(f(x))$ which a categorical distribution over $f(x) \in \mathcal{Y}_\Delta$.

Let $l : \mathcal{Y}_\Delta \times \mathcal{Y} \mapsto \mathbb{R}$ be a loss function, where $l\left(\hat{f}(x), y\right)$ with $\hat{f}(x) \in \mathcal{Y}_\Delta$ and $y \in \mathcal{Y}$ specifies the loss (e.g., cross-entropy, Hinge, L1, or L2 loss) to assign a data sample $x$ to the class $y$ by the hypothesis $\hat{f}$. Moreover, given a prediction probability $\hat{f}(x)$ w.r.t. the ground-truth prediction $f(x)$, we define the loss $\ell\left(\hat{f}(x), f(x)\right) = \mathbb{E}_{y \sim f(x)}\left[l\left(\hat{f}(x), y\right)\right] = \sum_{y=1}^C l\left(\hat{f}(x), y\right) f(x, y)$. This means $\ell(\cdot, \cdot)$ is convex w.r.t. the second argument. We further define the general loss caused by using a classifier $\hat{f} : \mathcal{X} \mapsto \mathcal{Y}_\Delta$ to predict $\mathbb{D} \equiv (\mathbb{P}, f)$ as

$$\mathcal{L}\left(\hat{f}, f, \mathbb{P}\right) = \mathcal{L}\left(\hat{f}, \mathbb{D}\right) := \mathbb{E}_{x \sim \mathbb{P}}\left[\ell\left(\hat{f}(x), f(x)\right)\right].$$

We inspect the multiple source setting in which we are given multiple source domains $\{\mathbb{D}^{S,i}\}_{i=1}^K$ over the common data space $\mathcal{X}$, each of which consists of data distribution and its own labeling

function $\mathbb{D}^{S,i} := (\mathbb{P}^{S,i}, f^{S,i})$. Based on these source domains, we need to work out a learner or classifier that requires to evaluate on a target domain $\mathbb{D}^T := (\mathbb{P}^T, f^T)$. Depending on the *knownness* or *unknownness* of a target domain during training, we experience either *multiple source DA* (MSDA) [31, 29, 42, 18] or *domain generalization* (DG) [24, 28, 27, 34] setting.

One typical approach in MSDA and DG is to combine the source domains together [14, 28, 27, 34, 1, 11, 46] to learn *DI representations* with hope to generalize well to a target domain. When combining the source domains, we obtain a mixture of multiple source distributions denoted as $\mathbb{D}^\pi = \sum_{i=1}^K \pi_i \mathbb{D}^{S,i}$, where the mixing coefficients $\pi = [\pi_i]_{i=1}^K$ can be conveniently set to $\pi_i = \frac{N_i}{\sum_{j=1}^K N_j}$ with $N_i$ being the training size of the $i$−th source domain.

In term of representation learning, input is mapped into a latent space $\mathcal{Z}$ by a feature map $g : \mathcal{X} \mapsto \mathcal{Z}$ and then a classifier $\hat{h} : \mathcal{Z} \mapsto \mathcal{Y}_\Delta$ is trained based on the representations $g(\mathcal{X})$. Let $f : \mathcal{X} \mapsto \mathcal{Y}_\Delta$ be the original labeling function. To facilitate the theory developed for latent space, we introduce representation distribution being the pushed-forward distribution $\mathbb{P}_g := g_\# \mathbb{P}$, and the labeling function $h : \mathcal{Z} \mapsto \mathcal{Y}_\Delta$ induced by $g$ as $h(z) = \frac{\int_{g^{-1}(z)} f(x)p(x)dx}{\int_{g^{-1}(z)} p(x)dx}$ [21]. Going back to our multiple source setting, the source mixture becomes $\mathbb{D}_g^\pi = \sum_i \pi_i \mathbb{D}_g^{S,i}$, where each source domain is $\mathbb{D}_g^{S,i} = (\mathbb{P}_g^{S,i}, h^{S,i})$, and the target domain is $\mathbb{D}_g^T = (\mathbb{P}_g^T, h^T)$.

Finally, in our theory development, we use Hellinger divergence between two distributions defined as $D_{1/2}(\mathbb{P}^1, \mathbb{P}^2) = 2 \int \left( \sqrt{p^1(x)} - \sqrt{p^2(x)} \right)^2 dx$, whose squared $d_{1/2} = \sqrt{D_{1/2}}$ is a proper metric.

## 2.2 Two Types of Domain-Invariant Representations

Hinted by the theoretical bounds developed in [3], *DI representations learning*, in which feature extractor $g$ maps source and target data distributions to a common distribution on the latent space, is well-grounded for the DA setting. However, this task becomes significantly challenging for the MSDA and DG settings due to multiple source domains and potential unknownness of target domain. Similar to the case of DA, it is desirable to develop theoretical bounds that directly motivate definitions of DI representations for the MSDA and DG settings, as being done in the next theorem.

**Theorem 1.** *Consider a mixture of source domains $\mathbb{D}^\pi = \sum_{i=1}^K \pi_i \mathbb{D}^{S,i}$ and the target domain $\mathbb{D}^T$. Let $\ell$ be any loss function upper-bounded by a positive constant $L$. For any hypothesis $\hat{f} : \mathcal{X} \mapsto \mathcal{Y}_\Delta$ where $\hat{f} = \hat{h} \circ g$ with $g : \mathcal{X} \to \mathcal{Z}$ and $\hat{h} : \mathcal{Z} \to \mathcal{Y}_\Delta$, the target loss on input space is upper bounded*

$$\mathcal{L}\left(\hat{f}, \mathbb{D}^T\right) \leq \sum_{i=1}^K \pi_i \mathcal{L}\left(\hat{f}, \mathbb{D}^{S,i}\right) + L \max_{i \in [K]} \mathbb{E}_{\mathbb{P}^{S,i}}\left[\|\Delta p^i(y|x)\|_1\right] + L\sqrt{2}\, d_{1/2}\left(\mathbb{P}_g^T, \mathbb{P}_g^\pi\right) \quad (1)$$

*where $\Delta p^i(y|x) := \left[\left|f^T(x,y) - f^{S,i}(x,y)\right|\right]_{y=1}^C$ is the absolute of single point label shift on input space between source domain $\mathbb{D}^{S,i}$, the target domain $\mathbb{D}^T$, and $[K] := \{1, 2, ..., K\}$.*

The bound in Equation 1 implies that the target loss in the input or latent space depends on three terms: (i) *representation discrepancy*: $d_{1/2}\left(\mathbb{P}_g^T, \mathbb{P}_g^\pi\right)$, (ii) *the label shift*: $\max_{i \in [K]} \mathbb{E}_{\mathbb{P}^{S,i}}\left[\|\Delta p^i(y|x)\|_1\right]$, and (iii) *the general source loss*: $\sum_{i=1}^K \pi_i \mathcal{L}\left(\hat{f}, \mathbb{D}^{S,i}\right)$. To minimize the target loss in the left side, we need to minimize the three aforementioned terms. First, the *label shift* term is a natural characteristics of domains, hence almost impossible to tackle. Secondly, the *representation discrepancy* term can be explicitly tackled for the MSDA setting, while almost impossible for the DG setting. Finally, the *general source loss* term is convenient to tackle, where its minimization results in a feature extractor $g$ and a classifier $\hat{h}$.

Contrary to previous works in DA and MSDA [3, 31, 54, 7] that consider both losses and data discrepancy on data space, our bound connects losses on data space to discrepancy on representation space. Therefore, our theory provides a natural way to analyse representation learning, especially feature alignment in deep learning practice. Note that although DANN [14] explains their feature alignment method using theory developed by Ben-david et al. [3], it is not rigorous. In particular, while application of the theory to representation space yield a representation discrepancy term, the loss terms are also on that feature space, and hence minimizing these losses is not the learning goal.

Finally, our setting is much more general, which extends to multilabel, stochastic labeling setting, and any bounded loss function.

From the upper bound, we turn to first type of DI representations for the MSDA and DG settings. Here, the objective of $g$ is to map samples onto representation space in a way that the common classifier $\hat{h}$ can effectively and correctly classifies them, regardless of which domain the data comes from.

**Definition 2.** *Consider a class $\mathcal{G}$ of feature maps and a class $\mathcal{H}$ of hypotheses. Let $\{\mathbb{D}^{S,i}\}_{i=1}^{K}$ be a set of source domains.*

*i) (**DG with unknown target data**) A feature map $g^* \in \mathcal{G}$ is said to be a **DG** general domain-invariant (**DI**) feature map if $g^*$ is the solution of the optimization problem (OP): $\min_{g \in \mathcal{G}} \min_{\hat{h} \in \mathcal{H}} \sum_{i=1}^{K} \pi_i \mathcal{L}\left(\hat{h}, \mathbb{D}_g^{S,i}\right)$. Moreover, the latent representations $z = g^*(x)$ induced by $g^*$ is called general DI representations for the DG setting.*

*ii) (**MSDA with known target data**) A feature map $g^* \in \mathcal{G}$ is said to be a **MSDA** general DI feature map if $g^*$ is the solution of the optimization problem (OP): $\min_{g \in \mathcal{G}} \min_{\hat{h} \in \mathcal{H}} \sum_{i=1}^{K} \pi_i \mathcal{L}\left(\hat{h}, \mathbb{D}_g^{S,i}\right)$ which satisfies $\mathbb{P}_{g^*}^T = \mathbb{P}_{g^*}^\pi$ (i.e., $\min_{g \in \mathcal{G}} d_{1/2}\left(\mathbb{P}_g^T, \mathbb{P}_g^\pi\right)$). Moreover, the latent representations $z = g^*(x)$ induced by $g^*$ is called general DI representations for the MSDA setting.*

The definition of general DI representations for the DG setting in Definition 2 is transparent in light of Theorem 1, wherein we aim to find $g$ and $\hat{h}$ to minimize the general source loss $\sum_{i=1}^{K} \pi_i \mathcal{L}\left(\hat{f}, \mathbb{D}^{S,i}\right)$ due to the unknownness of $\mathbb{P}^T$. Meanwhile, regarding the general DI representations for the MSDA setting, we aim to find $g^*$ satisfying $\mathbb{P}_{g^*}^T = \mathbb{P}_{g^*}^\pi$ and $\hat{h}$ to minimize the general source loss $\sum_{i=1}^{K} \pi_i \mathcal{L}\left(\hat{f}, \mathbb{D}^{S,i}\right)$. Practically, to find general representations for MSDA, we solve

$$
\min_{g \in \mathcal{G}} \min_{\hat{h} \in \mathcal{H}} \left\{ \sum_{i=1}^{K} \pi_i \mathcal{L}\left(\hat{h}, \mathbb{D}_g^{S,i}\right) + \lambda D\left(\mathbb{P}_g^T, \mathbb{P}_g^\pi\right) \right\},
$$

where $\lambda > 0$ is a trade-off parameter and $D$ can be any divergence (e.g., Jensen Shannon (JS) divergence [16], $f$-divergence [38], MMD distance [27], or WS distance [42]).

In addition, the general DI representations have been exploited in some works [57, 26] from the practical perspective and previously discussed in [2] from the theoretical perspective. Despite the similar definition to our work, general DI representation in [2] is not motivated from minimization of a target loss bound.

As pointed out in the next section, learning general DI representations increases the span of latent representations $g(x)$ on the latent space which might help to reduce $d_{1/2}\left(\mathbb{P}_g^T, \mathbb{P}_g^\pi\right)$ for a general target domain. On the other hand, many other works [27, 28, 15, 49, 11] have also explored the possibility of enhancing generalization by finding common representation among source distributions. The following theorem motivates the latter, which is the second kind of DI representations.

**Theorem 3.** *Consider a mixture of source domains $\mathbb{D}^\pi = \sum_{i=1}^{K} \pi_i \mathbb{D}^{S,i}$ and the target domain $\mathbb{D}^T$. Let $\ell$ be any loss function upper-bounded by a positive constant $L$. For any hypothesis $\hat{f} : \mathcal{X} \mapsto \mathcal{Y}_\Delta$ where $\hat{f} = \hat{h} \circ g$ with $g : \mathcal{X} \to \mathcal{Z}$ and $\hat{h} : \mathcal{Z} \to \mathcal{Y}_\Delta$, the target loss on input space is upper bounded*

$$
\mathcal{L}\left(\hat{f}, \mathbb{D}^T\right) \leq \sum_{i=1}^{K} \pi_i \mathcal{L}\left(\hat{f}, \mathbb{D}^{S,i}\right) + L \max_{i \in [K]} \mathbb{E}_{\mathbb{P}^{S,i}}\left[\|\Delta p^i(y|x)\|_1\right]
$$
$$
+ \sum_{i=1}^{K} \sum_{j=1}^{K} \frac{L\sqrt{2\pi_j}}{K} d_{1/2}\left(\mathbb{P}_g^T, \mathbb{P}_g^{S,i}\right) + \sum_{i=1}^{K} \sum_{j=1}^{K} \frac{L\sqrt{2\pi_j}}{K} d_{1/2}\left(\mathbb{P}_g^{S,i}, \mathbb{P}_g^{S,j}\right). \tag{2}
$$

Evidently, Theorem 3 suggests another kind of representation learning, where source representation distributions are aligned in order to lower the target loss's bound as concretely defined as follows.

**Definition 4.** *Consider a class $\mathcal{G}$ of feature maps and a class $\mathcal{H}$ of hypotheses. Let $\left\{\mathbb{D}^{S,i}\right\}_{i=1}^{K}$ be a set of source domains.*

*i) (**DG with unknown target data**) A feature map $g^* \in \mathcal{G}$ is a **DG** compressed DI representations for source domains $\{\mathbb{D}^{S,i}\}_{i=1}^K$ if $g^*$ is the solution of the optimization problem (OP):* $\min_{g \in \mathcal{G}} \min_{\hat{h} \in \mathcal{H}} \sum_{i=1}^K \pi_i \mathcal{L}\left(\hat{h}, \mathbb{D}_g^{S,i}\right)$ *which satisfies* $\mathbb{P}_{g^*}^{S,1} = \mathbb{P}_{g^*}^{S,2} = \ldots = \mathbb{P}_{g^*}^{S,K}$ *(i.e., the pushed forward distributions of all **source** domains are identical). The latent representations* $z = g^*(x)$ *is then called compressed DI representations for the DG setting.*

*ii) (**MSDA with known target data**) A feature map $g^* \in \mathcal{G}$ is an **MSDA** compressed DI representations for source domains $\{\mathbb{D}^{S,i}\}_{i=1}^K$ if $g^*$ is the solution of the optimization problem (OP):* $\min_{g \in \mathcal{G}} \min_{\hat{h} \in \mathcal{H}} \sum_{i=1}^K \pi_i \mathcal{L}\left(\hat{h}, \mathbb{D}_g^{S,i}\right)$ *which satisfies* $\mathbb{P}_{g^*}^{S,1} = \mathbb{P}_{g^*}^{S,2} = \ldots = \mathbb{P}_{g^*}^{S,K} = \mathbb{P}_{g^*}^T$ *(i.e., the pushed forward distributions of all **source** and **target** domains are identical). The latent representations* $z = g^*(x)$ *is then called compressed DI representations for the MSDA setting.*

### 2.3 Characteristics of Domain-Invariant Representations

In the previous section, two kinds of DI representations are introduced, where each of them originates from minimization of different terms in the upper bound of target loss. In what follows, we examine and discuss their benefits and drawbacks. We start with the novel development of *hypothesis-aware divergence for multiple distributions* , which is necessary for our theory developed later.

#### 2.3.1 Hypothesis-Aware Divergence for Multiple Distributions

Let consider multiple distributions $\mathbb{Q}_1, ..., \mathbb{Q}_C$ on the same space, whose density functions are $q_1, ..., q_C$. We are given a mixture of these distributions $\mathbb{Q}^\alpha = \sum_{i=1}^C \alpha_i \mathbb{Q}_i$ with a mixing coefficient $\alpha \in \mathcal{Y}_\Delta$ and desire to measure a divergence between $\mathbb{Q}_1, ..., \mathbb{Q}_C$. One possible solution is employing a hypothesis from an infinite capacity hypothesis class $\mathcal{H}$ to identify which distribution the data come from. In particular, given $z \sim \mathbb{Q}^\alpha$, the function $\hat{h}(z, i), i \in [C]$ outputs the probability that $x \sim \mathbb{Q}_i$. Note that we use the notation $\hat{h}$ in this particular subsection to denote a domain classifier, while everywhere else in the paper $\hat{h}$ denotes label classifier. Let $l\left(\hat{h}(z), i\right) \in \mathbb{R}$ be the loss when classifying $z$ using $\hat{h}$ provided the ground-truth label $i \in [C]$. Our motivation is that if the distributions $\mathbb{Q}_1, ..., \mathbb{Q}_C$ are distant, it is easier to distinguish samples from them, hence the minimum classification loss $\min_{\hat{h} \in \mathcal{H}} \mathcal{L}_{\mathbb{Q}_{1:C}}^\alpha\left(\hat{h}\right)$ is much lower than in the case of clutching $\mathbb{Q}_1, ..., \mathbb{Q}_C$. Therefore, we develop the following theorem to connect the loss optimization problem with an f-divergence among $\mathbb{Q}_1, ..., \mathbb{Q}_C$. More discussions about hypothesis-aware divergence can be found in Appendix B of this paper and in [13].

**Theorem 5.** *Assuming the hypothesis class $\mathcal{H}$ has infinite capacity, we define the hypothesis-aware divergence for multiple distributions as*

$$D^\alpha(\mathbb{Q}_1, ..., \mathbb{Q}_C) = -\min_{\hat{h} \in \mathcal{H}} \mathcal{L}_{\mathbb{Q}_{1:C}}^\alpha\left(\hat{h}\right) + \mathcal{C}_{l,\alpha}, \tag{3}$$

*where $\mathcal{C}_{l,\alpha}$ depends only on the form of loss function $l$ and value of $\alpha$. This divergence is a proper f-divergence among $\mathbb{Q}_1, ..., \mathbb{Q}_C$ in the sense that $D^\alpha(\mathbb{Q}_1, ..., \mathbb{Q}_C) \geq 0, \forall \mathbb{Q}_1, ..., \mathbb{Q}_C$ and $\alpha \in \mathcal{Y}_\Delta$, and $D^\alpha(\mathbb{Q}_1, ..., \mathbb{Q}_C) = 0$ if $\mathbb{Q}_1 = ... = \mathbb{Q}_C$.*

#### 2.3.2 General Domain-Invariant Representation

As previously defined, the general DI feature map $g^*$ (cf. Definition 2) is the one that minimizes the total source loss

$$g^* = \arg\min_{g \in \mathcal{G}} \min_{\hat{h} \in \mathcal{H}} \sum_{i=1}^K \pi_i \mathcal{L}\left(\hat{h}, h^{S,i}, \mathbb{P}_g^i\right) = \text{argmin}_{g \in \mathcal{G}} \min_{\hat{h} \in \mathcal{H}} \sum_{i=1}^K \pi_i \mathcal{L}\left(\hat{h}, \mathbb{D}_g^{S,i}\right). \tag{4}$$

From the result of Theorem 5, we expect that the minimal loss $\min_{\hat{h} \in \mathcal{H}} \mathcal{L}\left(\hat{h}, h^{S,i}, \mathbb{P}_g^{S,i}\right)$ should be inversely proportional to the divergence between the class-conditionals $\mathbb{P}_g^{S,i,c}$. To generalize this result to the multi-source setting, we further define $\mathbb{Q}_g^{S,c} := \sum_{i=1}^K \frac{\pi_i \gamma_{i,c}}{\alpha_c} \mathbb{P}_g^{S,i,c}$ as the mixture of the class $c$ conditional distributions of the source domains on the latent space, where $\alpha_c = \sum_{j=1}^K \pi_j \gamma_{j,c}$

are the mixing coefficients, and $\gamma_{i,c} = \mathbb{P}^{S,i}(y=c)$ are label marginals. Then, the inner loop of the objective function in Eq. 4 can be viewed as training the optimal hypothesis $\hat{h} \in \mathcal{H}$ to distinguish the samples from $\mathbb{Q}_g^{S,c}, c \in [C]$ for a given feature map $g$. Therefore, by linking to the multi-divergence concept developed in Section 2.3.1, we achieve the following theorem.

**Theorem 6.** *Assume that $\mathcal{H}$ has infinite capacity, we have the following statements.*

*1. $D^\alpha\left(\mathbb{Q}_g^{s,1}, ..., \mathbb{Q}_g^{s,C}\right) = -\min_{\hat{h} \in \mathcal{H}} \sum_{i=1}^K \pi_i \mathcal{L}\left(\hat{h}, h^{S,i}, \mathbb{P}_g^{S,i}\right) + const$, where $\alpha = [\alpha_c]_{c \in [C]}$ is defined as above.*

*2. Finding the general DI feature map $g^*$ via the OP in Eq. 4 is equivalent to solving*

$$g^* = \underset{g \in \mathcal{G}}{argmax}\, D^\alpha\left(\mathbb{Q}_g^{s,1}, ..., \mathbb{Q}_g^{s,C}\right). \tag{5}$$

Theorem 6, especially its second claim, discloses that learning general DI representations maximally expands the coverage of latent representations of the source domains by maximizing $D^\alpha\left(\mathbb{Q}_g^{s,1}, ..., \mathbb{Q}_g^{s,C}\right)$. Hence, the span of source mixture $\mathbb{P}_g^\pi = \sum_{c=1}^C \alpha_c \mathbb{Q}_g^{s,c}$ is also increased. We believe this demonstrates one of the benefits of general DI representations because it implicitly enhance the chance to match a general *unseen target domains* in the DG setting by possibly reducing the source-target representation discrepancy term $d_{1/2}\left(\mathbb{P}_g^T, \mathbb{P}_g^\pi\right)$ (cf. Theorem 1). Additionally, in MSDA where we wish to minimize $d_{1/2}\left(\mathbb{P}_g^T, \mathbb{P}_g^\pi\right)$ explicitly, expanding source representation's coverage is also useful.

### 2.3.3 Compressed Domain-Invariant Representations

It is well-known that generalization gap between the true loss and its empirical estimation affects generalization capability of model [48, 45]. One way to close this gap is increasing sampling density. However, as hinted by our analysis, enforcing general DI representation learning tends to maximize the cross-domain class divergence, hence implicitly increasing the diversity of latent representations from different source domains. This renders learning a classifier $\hat{h}$ on top of those source representations harder due to scattering samples on representation space, leading to higher generalization gap between empirical and general losses. In contrast, compressed DI representations help making the task of learning $\hat{h}$ easier with a lower generalization gap by decreasing the diversity of latent representations from the different source domains, via enforcing $\mathbb{P}_{g^*}^{S,1} = \mathbb{P}_{g^*}^{S,2} = \ldots = \mathbb{P}_{g^*}^{S,K}$. We now develop rigorous theory to examine this observation.

Let the training set be $S = \{(z_i, y_i)\}_{i=1}^N$, consisting of independent pairs $(z_i, y_i)$ drawn from the source mixture, i.e., the sampling process starts with sampling domain index $k \sim Cat(\pi)$, then sampling $z \sim \mathbb{P}_g^{S,k}$ (i.e., $x \sim \mathbb{P}^{S,k}$ and $z = g(x)$), and finally assigning a label with $y \sim h^{S,k}(z)$ (i.e., $h^{S,k}(z)$ is a distribution over $[C]$). The empirical loss is defined as

$$\mathcal{L}\left(\hat{h}, S\right) = \frac{1}{\sum_{k=1}^K N_k} \sum_{k=1}^K \sum_{i=1}^{N_k} \ell\left(\hat{h}(z_i), h^{S,k}(z_i)\right), \tag{6}$$

where $N_k$, $k \in [C]$ is the number of samples drawn from the $k$-th source domain and $N = \sum_{k=1}^K N_k$.

Here, $\mathcal{L}\left(\hat{h}, S\right)$ is an unbiased estimation of the general loss $\mathcal{L}\left(\hat{h}, \mathbb{D}_g^\pi\right) = \sum_{k=1}^K \pi_k \mathcal{L}\left(\hat{h}, \mathbb{D}_g^{S,k}\right)$. To quantify the quality of the estimation, we investigate the upper bound of the *generalization gap* $\left|\mathcal{L}\left(\hat{h}, S\right) - \mathcal{L}\left(\hat{h}, \mathbb{D}_g^\pi\right)\right|$ with the confidence level $1 - \delta$.

**Theorem 7.** *For any confident level $\delta \in [0,1]$ over the choice of $S$, the estimation of loss is in the $\epsilon$-range of the true loss*

$$Pr\left(\left|\mathcal{L}\left(\hat{h}, S\right) - \mathcal{L}\left(\hat{h}, \mathbb{D}_g^\pi\right)\right| \le \epsilon\right) \ge 1 - \delta,$$

*where $\epsilon = \epsilon(\delta) = \left(\frac{A}{\delta}\right)^{1/2}$ is a function of $\delta$ for which $A$ is proportional to*

$$\frac{1}{N}\left(\sum_{i=1}^K \sum_{j=1}^K \frac{\sqrt{\pi_i}}{K} \mathcal{L}\left(\hat{f}, \mathbb{D}^{S,j}\right) + L \sum_{i=1}^K \sqrt{\pi_i} \max_{k \in [K]} \mathbb{E}_{\mathbb{P}^{S,k}}\left[\left\|\Delta p^{k,i}(y|x)\right\|_1\right] + \frac{L}{K} \sum_{i=1}^K \sum_{j=1}^K \sqrt{2\pi_i}\, d_{1/2}\left(\mathbb{P}_g^{S,i}, \mathbb{P}_g^{S,j}\right)\right)^2$$

Theorem 7 reveals one benefit of learning compressed DI representations, that is, when enforcing compressed DI representation learning, we minimize $\frac{L}{K} \sum_{i=1}^{K} \sum_{j=1}^{K} \sqrt{2\pi_i} \, d_{1/2} \left( \mathbb{P}_g^{S,i}, \mathbb{P}_g^{S,j} \right)$, which tends to reduce the generalization gap $\epsilon = \epsilon(\delta)$ for a given confidence level $1 - \delta$. Therefore, compressed DI representations allow us to minimize population loss $\mathcal{L} \left( \hat{h}, \mathbb{D}_g^{\pi} \right)$ more efficiently via minimizing empirical loss.

## 2.4 Trade-off of Learning Domain Invariant Representation

Similar to the theoretical finding in Zhao et al. [53] developed for DA, we theoretically find that compression does come with a cost for MSDA and DG. We investigate the representation trade-off, typically how compressed DI representation affects classification loss. Specifically, we consider a data processing chain $\mathcal{X} \xmapsto{g} \mathcal{Z} \xmapsto{\hat{h}} \mathcal{Y}_{\Delta}$, where $\mathcal{X}$ is the common data space, $\mathcal{Z}$ is the latent space induced by a feature extractor $g$, and $\hat{h}$ is a hypothesis on top of the latent space. We define $\mathbb{P}_{\mathcal{Y}}^{\pi}$ and $\mathbb{P}_{\mathcal{Y}}^{T}$ as two distribution over $\mathcal{Y}$ in which to draw $y \sim \mathbb{P}_{\mathcal{Y}}^{\pi}$, we sample $k \sim Cat(\pi)$, $x \sim \mathbb{P}^{S,k}$, and $y \sim f^{S,k}(x)$, while similar to draw $y \sim \mathbb{P}_{\mathcal{Y}}^{T}$. Our theoretical bounds developed regarding the trade-off of learning DI representations are relevant to $d_{1/2} \left( \mathbb{P}_{\mathcal{Y}}^{\pi}, \mathbb{P}_{\mathcal{Y}}^{T} \right)$.

**Theorem 8.** *Consider a feature extractor $g$ and a hypothesis $\hat{h}$, the Hellinger distance between two label marginal distributions $\mathbb{P}_{\mathcal{Y}}^{\pi}$ and $\mathbb{P}_{\mathcal{Y}}^{T}$ can be upper-bounded as:*

*1. $d_{1/2} \left( \mathbb{P}_{\mathcal{Y}}^{\pi}, \mathbb{P}_{\mathcal{Y}}^{T} \right) \leq \left[ \sum_{k=1}^{K} \pi_k \mathcal{L} \left( \hat{h} \circ g, f^{S,k}, \mathbb{P}^{S,k} \right) \right]^{1/2} + d_{1/2} \left( \mathbb{P}_g^{T}, \mathbb{P}_g^{\pi} \right) + \mathcal{L} \left( \hat{h} \circ g, f^{T}, \mathbb{P}^{T} \right)^{1/2}$*

*2. $d_{1/2} \left( \mathbb{P}_{\mathcal{Y}}^{\pi}, \mathbb{P}_{\mathcal{Y}}^{T} \right) \leq \left[ \sum_{i=1}^{K} \pi_i \mathcal{L} \left( \hat{h} \circ g, f^{S,i}, \mathbb{P}^{S,i} \right) \right]^{1/2} + \sum_{i=1}^{K} \sum_{j=1}^{K} \frac{\sqrt{\pi_j}}{K} d_{1/2} \left( \mathbb{P}_g^{S,i}, \mathbb{P}_g^{S,j} \right) + \sum_{i=1}^{K} \sum_{j=1}^{K} \frac{\sqrt{\pi_j}}{K} d_{1/2} \left( \mathbb{P}_g^{T}, \mathbb{P}_g^{S,i} \right) + \mathcal{L} \left( \hat{h} \circ g, f^{T}, \mathbb{P}^{T} \right)^{1/2}$.*

*Here we note that the general loss $\mathcal{L}$ is defined based on the Hellinger loss $\ell$ which is define as*

$$\ell(\hat{f}(x), f(x)) = D_{1/2}(\hat{f}(x), f(x)) = 2 \sum_{i=1}^{C} \left( \sqrt{\hat{f}(x,i)} - \sqrt{f(x,i)} \right)^2 \text{ (more discussion can be}$$

*found in Appendix C).*

*Remark.* Compared to the trade-off bound in the work of Zhao et al. [53], our context is more general, concerning MSDA and DG problems with multiple source domains and multi-class probabilistic labeling functions, rather than single source DA with binary-class and deterministic setting. Moreover, the Hellinger distance is more universal, in the sense that it does not depend on the choice of classifier family $\mathcal{H}$ and loss function $\ell$ as in the case of $\mathcal{H}$-divergence in [53].

We base on the first inequality of Theorem 8 to analyze the trade-off of learning general DI representations. The first term on the left hand side is the source mixture's loss, which is controllable and tends to be small when enforcing learning general DI representations. With that in mind, if *two label marginal distributions* $\mathbb{P}_{\mathcal{Y}}^{\pi}$ and $\mathbb{P}_{\mathcal{Y}}^{T}$ are distant (i.e., $d_{1/2} \left( \mathbb{P}_{\mathcal{Y}}^{\pi}, \mathbb{P}_{\mathcal{Y}}^{T} \right)$ is high), the sum $d_{1/2} \left( \mathbb{P}_g^{T}, \mathbb{P}_g^{\pi} \right) + \mathcal{L} \left( \hat{h} \circ g, f^{T}, \mathbb{P}^{T} \right)^{1/2}$ tends to be high. This leads to 2 possibilities. The first scenario is when the representation discrepancy $d_{1/2} \left( \mathbb{P}_g^{T}, \mathbb{P}_g^{\pi} \right)$ has small value, e.g., it is minimized in MSDA setting, or it happens to be small by pure chance in DG setting. In this case, the lower bound of target loss $\mathcal{L} \left( \hat{h} \circ g, f^{T}, \mathbb{P}^{T} \right)$ is high, possibly hurting model's generalization ability. On the other hand, if the discrepancy $d_{1/2} \left( \mathbb{P}_g^{T}, \mathbb{P}_g^{\pi} \right)$ is large for some reasons, the lower bound of target loss will be small, but its upper-bound is higher, as indicated Theorem 1.

Based on the second inequality of Theorem 8, we observe that if *two label marginal distributions* $\mathbb{P}_{\mathcal{Y}}^{\pi}$ and $\mathbb{P}_{\mathcal{Y}}^{T}$ are distant while enforcing learning compressed DI representations (i.e., both source loss and source-source feature discrepancy $\left[ \sum_{i=1}^{K} \pi_i \mathcal{L} \left( \hat{h} \circ g, f^{S,i}, \mathbb{P}^{S,i} \right) \right]^{1/2} + \sum_{i=1}^{K} \sum_{j=1}^{K} \frac{\sqrt{\pi_j}}{K} d_{1/2} \left( \mathbb{P}_g^{S,i}, \mathbb{P}_g^{S,j} \right)$ are low), the sum $\sum_{i=1}^{K} \sum_{j=1}^{K} \frac{\sqrt{\pi_j}}{K} d_{1/2} \left( \mathbb{P}_g^{T}, \mathbb{P}_g^{S,i} \right) + \mathcal{L} \left( h \circ g, f^{T}, \mathbb{P}^{T} \right)^{1/2}$ is high. For the *MSDA setting*, the

discrepancy $\sum_{i=1}^{K} \sum_{j=1}^{K} \frac{\sqrt{\pi}_j}{K} d_{1/2} \left( \mathbb{P}_g^T, \mathbb{P}_g^{S,i} \right)$ is trained to get smaller, meaning that the lower bound of target loss $\mathcal{L} \left( \hat{h} \circ g, f^T, \mathbb{P}^T \right)$ is high, hurting the target performance. Similarly, for the *DG setting*, if the trained feature extractor $g$ occasionally reduces $\sum_{i=1}^{K} \sum_{j=1}^{K} \frac{\sqrt{\pi}_j}{K} d_{1/2} \left( \mathbb{P}_g^T, \mathbb{P}_g^{S,i} \right)$ for some unseen target domain, it certainly increases the target loss $\mathcal{L} \left( h \circ g, f^T, \mathbb{P}^T \right)$. In contrast, if for some target domains, the discrepancy $\sum_{i=1}^{K} \sum_{j=1}^{K} \frac{\sqrt{\pi}_j}{K} d_{1/2} \left( \mathbb{P}_g^T, \mathbb{P}_g^{S,i} \right)$ is high by some reasons, by linking to the upper bound in Theorem 3, the target general loss has a high upper-bound, hence is possibly high.

This trade-off between representation discrepancy and target loss suggests a sweet spot for just-right feature alignment. In that case, the target loss is most likely to be small.

# 3 Experiment

To investigate the trade-off of learning DI representations, we conduct domain generalization experiments on the *colored MNIST* dataset (CC BY-SA 3.0) [2, 23]. In particular, the task is to predict binary label $Y$ of colored input images $X$ generated from binary digit feature $Z_d$ and color feature $Z_c$. We refer readers to Appendix D for more information of this dataset.

We conduct 7 source domains by setting color-label correlation $\mathbb{P}(Z_c = 1|Y = 1) = \mathbb{P}(Z_c = 0|Y = 0) = \theta^{S,i}$ where $\theta^{s,i} \sim Uni([0.6, 1])$ for $i = 1, ..., 12$, while two target domains are created with $\theta^{T,i} \in \{0.05, 0.7\}$ for $i = 1, 2$. Here we note that colored images in the target domain with $\theta^{T,2} = 0.7$ are *more similar* to those in the source domains, while colored images in the target domain with $\theta^{T,1} = 0.05$ are *less similar*.

We wish to study the characteristics and trade-off between two kinds of DI representations when predicting on various target domains. Specifically, we apply adversarial learning [16] similar to [14], in which a min-max game is played between domain discriminator $\hat{h}^d$ trying to distinguish the source domain given representation, while the feature extractor (generator) $g$ tries to fool the domain discriminator. Simultaneously, a classifier is used to classify label based on the representation. Let $\mathcal{L}_{gen}$ and $\mathcal{L}_{disc}$ be the label classification and domain discrimination losses, the training objective becomes:

$$\min_g \left( \min_{\hat{h}} \mathcal{L}_{gen} + \lambda \max_{\hat{h}^d} \mathcal{L}_{disc} \right),$$

where the source compression strength $\lambda > 0$ controls the compression extent of learned representation. More specifically, general DI representation is obtained with $\lambda = 0$, while larger $\lambda$ leads to more compressed DI representation. Finally, our implementation is based on DomainBed [17] repository, and all training details as well as further MSDA experiment and DG experiment on real datasets are included in Appendix D.[1]

## 3.1 Trade-off of Two Kinds of Domain-Invariant Representations

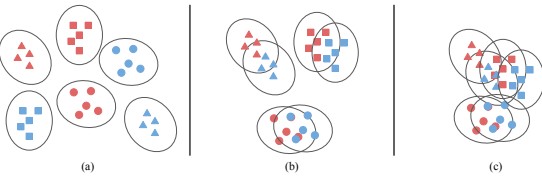

Figure 1: Phase transition of representations with increased $\lambda$. (a) General DI representations. (b) Just-right compressed DI representations. (c) Overly-compressed DI representations.

---

[1]Our code could be found at: https://github.com/VinAIResearch/DIRep

We consider target domain which is similar to source domain in this experiment. To govern the trade-off between two kinds of DI representations, we gradually increase the source compression strength $\lambda$ with noting that $\lambda = 0$ means only focusing on learning general DI representations. According to our theory and as shown in Figure 1, *learning only general DI representations* $(\lambda = 0)$ maximizes the cross-domain class divergence by encouraging the classes of source domains more separate arbitrarily, while by increasing $\lambda$, we enforce compressing the latent representations of source domains together. Therefore, for an appropriate $\lambda$, the class separation from general DI representations and the source compression from compressed DI representations (i.e., *just-right compressed DI representations*), are balanced as in the case (b) of Figure 1, while for overly high $\lambda$, source compression from compressed DI representations dominates and compromises the class separation from general DI representations (i.e., *overly-compressed DI representations*).

Figure 2a shows the source validation and target accuracies when increasing $\lambda$ (i.e., encouraging the source compression). It can be observed that both source validation accuracy and target accuracy have the same pattern: increasing when setting appropriate $\lambda$ for *just-right compressed DI representations* and compromising when setting overly high values $\lambda$ for *overly-compressed DI representations*. Figure 2b shows in detail the variation of the source validation accuracy for each specific $\lambda$. In practice, we should encourage learning two kinds of DI representations simultaneously by finding an appropriate trade-off to balance them for working out just-right compressed DI representations.

## 3.2 Generalization Capacity on Various Target Domains

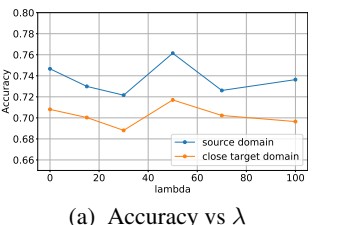

(a) Accuracy vs $\lambda$

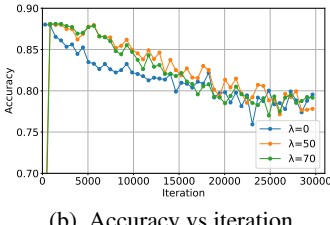

(b) Accuracy vs iteration

Figure 2: (**2a**) Source validation accuracy and target accuracy for close target domain (see Section 3.2) over compression strength. (**2b**) Validation accuracy over training step for different values of $\lambda$.

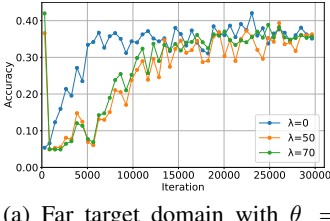

(a) Far target domain with $\theta = 0.05$.

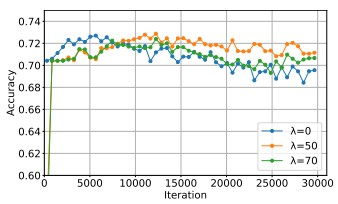

(b) Close target domain with $\theta = 0.7$.

Figure 3: Accuracy of models trained with different $\lambda$ on two test domains. For the far domain with $\theta = 0.05$, the model corresponding with general DI representations performs better than that with more compact DI representations. For close domain with $\theta = 0.7$, the models with compression generalize significantly better than those with general DI representations.

In this experiment, we inspect the generalization capacity of the models trained on the source domains with different compression strengths $\lambda$ to different target domains (i.e., close and far ones). As shown in Figure 3, the target accuracy is lower for the far target domain than for the close one regardless of the source compression strengths $\lambda$. This observation aligns with our upper-bounds developed in Theorems 1 and 3 in which larger representation discrepancy increases the upper-bounds of the general target loss, hence more likely hurting the target performance.

Another interesting observation is that for the far target domain, the target accuracy for $\lambda = 0$ peaks. This can be partly explained as the general DI representations help to increase the span of source latent representations for more chance to match a far target domain (cf. Section 2.3.2). Specifically, the source-target discrepancy $d_{1/2} \left( \mathbb{P}_g^T, \mathbb{P}_g^\pi \right)$ could be small and the upper bound in Theorem 1 is lower. In contrast, for the close target domain, the compressed DI representations help to really improve

the target performance, while over-compression degrades the target performance, similar to result on source domains. This is because the target domain is naturally close to the source domains, the source and target latent representations are already mixed up, but by encouraging the compressed DI representations, we aim to learning more elegant representations for improving target performance.

# 4 Related Works

*Domain adaptation (DA)* has been intensively studied from both theoretical [3, 9, 6, 53, 33, 43] and empirical [44, 52, 14, 7, 51] perspectives. Notably, the pioneering work [3] and other theoretical works [43, 52, 53, 21, 30, 6] have investigated DA in various aspects which lay foundation for the success of practical works [14, 47, 44, 52, 14, 7, 51]. *Multiple-source domain adaptation (MSDA)* extends DA by gathering training set from multiple source domains [8, 29, 20, 42, 43, 12, 54, 50, 40]. Different from DA with abundant theoretical works, theoretical study in MSDA is significantly limited. Notably, the works in [31, 32] relies on assumptions about the existence of the same labeling function, which can be used for all source domains; and target domain is an unknown mixture of the source domains. They then show that there exists a distributional weighted combination of these experts performing well on target domain with loss at most $\epsilon$, given a set of source expert hypotheses (with loss is at most on respective domain). Hoffman et al. [18] further develop this idea to a more general case where different labeling functions corresponding to different source domains, and the target domain is arbitrary. Under this setting, there exists a distributional weighted combination of source experts such that the loss on target domain is bounded by a source loss term and a discrepancy term between target domain and source mixture. On the other hand, Zhao et al. [54] directly extend Ben-david et al.'s work with an improvement on the sample complexity term.

*Domain generalization (DG)* [17, 25, 4, 19] is the most challenging setting among the three due to the unavailability of target data. The studies in [4, 10] use kernel method to find feature map which minimizes expected target loss. Moreover, those in [27, 28, 34, 15, 49, 11] learn compressed DI representations by minimizing different types of domain dissimilarity. Other notions of invariance are also proposed, for example, [2] learns a latent space on which representation distributions do not need to be aligned, but share a common optimal hypothesis. Another work in [41] uncovers domain-invariant hypothesis from low-rank decomposition of domain-specific hypotheses. Moreover, there are other works which try to strike a balance between two types of representation spaces [5].

# 5 Conclusion

In this paper, we derive theoretical bounds for target loss in MSDA and DG settings which characterize two types of representation: general DI representation for learning invariant classifier which works on all source domains and compressed DI representation motivated from reducing inter-domain representation discrepancy. We further characterize the properties of these two representations, and develop a lower bound on the target loss which governs the trade-off between learning them. Finally, we conduct experiments on Colored MNIST dataset and real dataset to illustrate our theoretical claims.

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
