# Appendix: On Learning Domain-Invariant Representations for Transfer Learning with Multiple Sources

**Trung Phung**[1] **Trung Le**[2] **Long Vuong**[1] **Toan Tran**[1] **Anh Tran**[1] **Hung Bui**[1] **Dinh Phung**[1,2]

[1] VinAI Research, Vietnam   [2] Monash University, Australia

`v.trungpq3@vinai.io, trunglm@monash.edu,`
`{v.longvt8, v.toantm3, v.anhtt152, v.hungbh1, v.dinhpq2}@vinai.io`

This supplementary material provides proofs for theoretical results stated in the main paper, as well as detailed experiment settings and further experimental results. It is organized as follows

- Appendix A contains the proofs for the upper bounds of target loss introduced in Section 2.2 of our main paper.

- Appendix B contains the proofs for characteristics of two representations as mentioned in Section 2.3 of our main paper.

- Appendix C contains the proof for trade-off theorem discussed in Section 2.4 of our main paper.

- Finally, in Appendix D, we present the generative detail of the Colored MNIST dataset, the experimental setting together with additional results for MSDA on Colored MNIST and DG on the real-world PACS dataset.

## 1 Appendix A: Target Loss's Upper Bounds

We begin with a crucial proposition for our theory development, which further allows us to connect loss on feature space to loss on data space.

**Proposition 1.** *Let* $\hat{f} : \mathcal{X} \mapsto \mathcal{Y}_\Delta$ *where* $\hat{f} = \hat{h} \circ g$ *with* $g : \mathcal{X} \mapsto \mathcal{Z}$ *and* $\hat{h} : \mathcal{Z} \mapsto \mathcal{Y}_\Delta$. *Let* $c : \mathcal{Z} \times [C] \mapsto \mathbb{R}$ *be a positive function.*

*i) For any* $i \in [C]$*, we have*

$$\int h\left(z, i\right) c\left(\hat{h}\left(z\right), i\right) p_g\left(z\right) dz = \int f\left(x, i\right) c\left(\hat{f}\left(x\right), i\right) p\left(x\right) dx,$$

*ii) For any* $i \in [C]$*, we have*

$$\int \left|h\left(z, i\right) - h'\left(z, i\right)\right| c\left(\hat{h}\left(z\right), i\right) p_g\left(z\right) dz \leq \int \left|f\left(x, i\right) - f'\left(x, i\right)\right| c\left(\hat{f}\left(x\right), i\right) p\left(x\right) dx,$$

*where* $p$ *is the density of a distribution* $\mathbb{P}$ *on* $\mathcal{X}$*,* $p_g$ *is the density of the distribution* $\mathbb{P}_g = g_\# \mathbb{P}$*,* $f : \mathcal{X} \mapsto \mathcal{Y}_\Delta$ *and* $f' : \mathcal{X} \mapsto \mathcal{Y}_\Delta$ *are labeling functions, and* $h : \mathcal{Z} \mapsto \mathcal{Y}_\Delta$ *and* $h' : \mathcal{Z} \mapsto \mathcal{Y}_\Delta$ *are labeling functions on the latent space induced from* $f$*, i.e.,* $h(z) = \frac{\int_{g^{-1}(z)} f(x,i) p(x) dx}{\int_{g^{-1}(z)} p(x) dx}$ *and* $h'(z) = \frac{\int_{g^{-1}(z)} f'(x,i) p(x) dx}{\int_{g^{-1}(z)} p(x) dx}$.

*Proof.* i) Using the definition of $h$, we manipulate the integral on feature space as

$$\int_{\mathcal{Z}} h(z,i) c\left(\hat{h}(z),i\right) p_g(z) dz \overset{(1)}{=} \int_{\mathcal{Z}} c\left(\hat{h}(z),i\right) \frac{\int_{g^{-1}(z)} f(x,i) p(x) dx}{\int_{g^{-1}(z)} p(x) dx} \int_{g^{-1}(z)} p(x') dx' dz$$

$$= \int_{\mathcal{Z}} c\left(\hat{h}(z),i\right) \int_{g^{-1}(z)} f(x,i) p(x) dx dz$$

$$\overset{(2)}{=} \int_{\mathcal{Z}} \int_{g^{-1}(z)} c\left(\hat{h}(g(x)),i\right) f(x,i) p(x) dx dz$$

$$\overset{(3)}{=} \int_{\mathcal{Z}} \int_{\mathcal{X}} \mathbb{I}_{x \in g^{-1}(z)} c\left(\hat{h}(z),i\right) f(x,i) p(x) dx dz$$

$$\overset{(4)}{=} \int_{\mathcal{X}} \int_{\mathcal{Z}} \mathbb{I}_{z=g(x)} c\left(\hat{h}(z),i\right) f(x,i) p(x) dx dz$$

$$= \int_{\mathcal{X}} c\left(\hat{h}(g(x)),i\right) f(x,i) p(x) dx$$

$$= \int_{\mathcal{X}} c\left(\hat{f}(x),i\right) f(x,i) p(x) dx.$$

In (1), we use the definition of push-forward distribution $\int_B p_g(z) dz = \int_B dz \int_{g^{-1}(z)} p(x) dx$. In (2), $c\left(\hat{h}(z),i\right)$ can be put inside the integral because $z = g(x)$ for any $x \in g^{-1}(z)$. In (3), the integral over restricted region is expanded to all space with the help of $\mathbb{I}_{x \in g^{-1}(z)}$, whose value is 1 if $x \in g^{-1}(z)$ and 0 otherwise. In (4), Fubini theorem is invoked to swap the integral signs.

ii) Using the same technique, we have

$$\int \left| h(z,i) - h'(z,i) \right| c\left(\hat{h}(z),i\right) p_g(z) dz$$

$$= \int_{\mathcal{Z}} c\left(\hat{h}(z),i\right) \frac{\left| \int_{g^{-1}(z)} f(x,i) p(x) dx - \int_{g^{-1}(z)} f'(x,i) p(x) dx \right|}{\int_{g^{-1}(z)} p(x') dx'} \int_{g^{-1}(z)} p(x) dx dz$$

$$= \int_{\mathcal{Z}} c\left(\hat{h}(z),i\right) \left| \int_{g^{-1}(z)} (f(x,i) - f'(x,i)) p(x) dx \right| dz$$

$$\leq \int_{\mathcal{Z}} c\left(\hat{h}(z),i\right) \int_{g^{-1}(z)} \left| f(x,i) - f'(x,i) \right| p(x) dx dz$$

$$= \int_{\mathcal{Z}} \int_{\mathcal{X}} \mathbb{I}_{x \in g^{-1}(z)} c\left(\hat{h}(z),i\right) \left| f(x,i) - f'(x,i) \right| p(x) dx dz$$

$$= \int_{\mathcal{X}} \int_{\mathcal{Z}} \mathbb{I}_{z=g(x)} c\left(\hat{h}(g(x)),i\right) \left| f(x,i) - f'(x,i) \right| p(x) dx dz$$

$$= \int_{\mathcal{X}} \left| f(x,i) - f'(x,i) \right| c\left(\hat{f}(x),i\right) p(x) dx.$$

$\square$

Proposition 1 allows us to connect expected loss on feature space and input space as in the following corollary.

**Corollary 2.** *Consider a domain $\mathbb{D} = (\mathbb{P}, f)$ with data distribution $\mathbb{P}$ and ground-truth labeling function $f$. A hypothesis is $\hat{f} : \mathcal{X} \mapsto \mathcal{Y}_\Delta$, where $\hat{f} = \hat{h} \circ g$ with $g : \mathcal{X} \mapsto \mathcal{Z}$ and $\hat{h} : \mathcal{Z} \mapsto \mathcal{Y}_\Delta$. Then, the labeling function $f$ on input space induces a ground-truth labeling function on feature space $h(z) = \frac{\int_{g^{-1}(z)} f(x,i) p(x) dx}{\int_{g^{-1}(z)} p(x) dx}$. Let $\ell : \mathcal{Y}_\Delta \times \mathcal{Y}_\Delta \mapsto \mathbb{R}$ be the loss function, then expected loss can be calculated either w.r.t. input space $\mathcal{L}\left(\hat{f}, f, \mathbb{P}\right) = \int \ell\left(\hat{f}(x), f(x)\right) p(x) dx$ or w.r.t. feature*

*space* $\mathcal{L}\left(\hat{h}, h, g_{\#}\mathbb{P}\right) = \int \ell\left(\hat{h}(z), h(z)\right) p_g(z)\, dz$. *If we assume the loss* $\ell(\cdot, \cdot)$ *has the formed mentioned in the main paper, that is,*

$$\ell(u, v) = \sum_{i=1}^{C} l(u, i)\, v_i,$$

*for any two simplexes* $u, v \in \mathcal{Y}_{\Delta}$, *where* $u = [u_i]_{i=1}^{C}$ *and* $v = [v_i]_{i=1}^{C}$. *Then the losses w.r.t. input space and feature space are the same, i.e.,*

$$\mathcal{L}\left(\hat{h}, h, g_{\#}\mathbb{P}\right) = \mathcal{L}\left(\hat{f}, f, \mathbb{P}\right).$$

*Proof.* We derive as

$$\mathcal{L}\left(\hat{h}, h, g_{\#}\mathbb{P}\right) = \int_{\mathcal{Z}} \ell\left(\hat{h}(z), h(z)\right) p(z)\, dz = \sum_{i=1}^{C} \int_{\mathcal{Z}} l\left(\hat{h}(z), i\right) h(z, i)\, p(z)\, dz$$

$$\overset{(1)}{=} \sum_{i=1}^{C} \int_{\mathcal{X}} l\left(\hat{f}(x), i\right) f(x, i)\, p(x)\, dx = \int_{\mathcal{X}} \ell\left(\hat{f}(x), f(x)\right) p(x)\, dx$$

$$= \mathcal{L}\left(\hat{f}, f, \mathbb{P}\right).$$

Here we have $\overset{(1)}{=}$ by using (i) in Proposition 1 with $c(\beta, i) = l(\beta, i)$ for any $\beta \in \mathcal{Y}_{\Delta}$ and $i \in [C]$.  □

Normally, target loss $\mathcal{L}\left(\hat{f}, f^T, \mathbb{P}^T\right)$ is bounded by source loss $\mathcal{L}\left(\hat{f}, f^S, \mathbb{P}^S\right)$, a label shift term $LS\left(f^T, f^S\right)$, and a data shift term $DS\left(\mathbb{P}^T, \mathbb{P}^S\right)$ [2]. Here, this kind of bound is developed using data distribution $\mathbb{P}$ on input space and labeling function $f$ from input to label space, which are not convenient in understanding representation learning, since $\mathbb{P}^T, \mathbb{P}^S$ are data nature and therefore fixed. In order to relate target loss to properties of learned representations, another bound in which $\mathcal{L}\left(\hat{h}, h, \mathbb{P}_g\right)$ is the loss w.r.t. feature space and $DS\left(\mathbb{P}_g^T, \mathbb{P}_g^S\right)$ is the data shift on feature space is more favorable. However, this naive approach presents a pitfall, since the loss $\mathcal{L}\left(\hat{h}, h, \mathbb{P}_g\right)$ is not identical to the loss w.r.t. input space $\mathcal{L}\left(\hat{f}, f, \mathbb{P}\right)$, which is of ultimate interest, e.g., to-be-bounded target loss $\mathcal{L}\left(\hat{f}, f^T, \mathbb{P}^T\right)$, or to-be-minimized source loss $\mathcal{L}\left(\hat{f}, f^S, \mathbb{P}^S\right)$. Using the previous proposition and corollary, we could bridge this gap and develop a target bound connecting both data space and feature space.

**Theorem 3.** (**Theorem 1 in the main paper**) *Consider a mixture of source domains* $\mathbb{D}^{\pi} = \sum_{i=1}^{K} \pi_i \mathbb{D}^{S,i}$ *and the target domain* $\mathbb{D}^T$. *Let* $\ell$ *be any loss function upper-bounded by a positive constant* $L$. *For any hypothesis* $\hat{f} : \mathcal{X} \mapsto \mathcal{Y}_{\Delta}$ *where* $\hat{f} = \hat{h} \circ g$ *with* $g : \mathcal{X} \mapsto \mathcal{Z}$ *and* $\hat{h} : \mathcal{Z} \mapsto \mathcal{Y}_{\Delta}$, *the target loss on input space is upper bounded*

$$\mathcal{L}\left(\hat{f}, \mathbb{D}^T\right) \leq \sum_{i=1}^{K} \pi_i \mathcal{L}\left(\hat{f}, \mathbb{D}^{S,i}\right) + L \max_{i \in [K]} \mathbb{E}_{\mathbb{P}^{S,i}}\left[\|\Delta p^i(y|x)\|_1\right] + L\sqrt{2}\, d_{1/2}\left(\mathbb{P}_g^T, \mathbb{P}_g^{\pi}\right), \quad (1)$$

*where* $\Delta p^i(y|x) := \left[\left|f^T(x, y) - f^{S,i}(x, y)\right|\right]_{y=1}^{C}$ *is the absolute of single point label shift on input space between source domain* $\mathbb{D}^{S,i}$, *the target domain* $\mathbb{D}^T$, $[K] := \{1, 2, ..., K\}$, *and the **feature distribution of the source mixture*** $\mathbb{P}_g^{\pi} := \sum_{i=1}^{K} \pi_i \mathbb{P}_g^{S,i}$.

*Proof.* First, consider the hybrid domain $\mathbb{D}_g^{h,i} = \left(\mathbb{P}_g^{\pi}, h^T\right)$, with $\mathbb{P}_g^{\pi} := \sum_{i=1}^{K} \pi_i \mathbb{P}_g^{S,i}$ be the feature distribution of the source mixture, and $h^T$ is the induced ground-truth labeling function of target

domain. The loss on the hybrid domain is then upper bounded by the loss on source mixture and a label shift term. We derive as follows:

$$\mathcal{L}\left(\hat{h}, \mathbb{D}_g^{hi}\right) = \int \ell\left(\hat{h}\left(z\right), h^T\left(z\right)\right) p_g^\pi\left(z\right) dz$$

$$= \int \ell\left(\hat{h}\left(z\right), h^T\left(z\right)\right) \sum_{i=1}^K \pi_i p_g^{S,i}\left(z\right) dz = \sum_{i=1}^K \pi_i \int \ell\left(\hat{h}\left(z\right), h^T\left(z\right)\right) p_g^{S,i}\left(z\right) dz$$

$$\leq \sum_{i=1}^K \pi_i \int \ell\left(\hat{h}\left(z\right), h^{S,i}\left(z\right)\right) p_g^{S,i}\left(z\right) dz$$

$$+ \sum_{i=1}^K \pi_i \int \left|\ell\left(\hat{h}\left(z\right), h^T\left(z\right)\right) - \ell\left(\hat{h}\left(z\right), h^{S,i}\left(z\right)\right)\right| p_g^{S,i}\left(z\right) dz.$$

Firstly, using Corollary 2, the loss terms on feature space and input space are equal

$$\sum_{i=1}^K \pi_i \int \ell\left(\hat{h}\left(z\right), h^{S,i}\left(z\right)\right) p_g^{S,i}\left(z\right) dz = \sum_{i=1}^K \pi_i \mathcal{L}\left(\hat{f}, f^{S,i}, \mathbb{P}^{S,i}\right)$$

Secondly, the difference term, can be transformed into label shift on input space using Proposition 1

$$\sum_{i=1}^K \pi_i \int_{\mathcal{Z}} \left|\ell\left(\hat{h}\left(z\right), h^T\left(z\right)\right) - \ell\left(\hat{h}\left(z\right), h^{S,i}\left(z\right)\right)\right| p_g^{S,i}\left(z\right) dz$$

$$\leq \sum_{i=1}^K \pi_i \sum_{j=1}^C \int_{\mathcal{Z}} \ell\left(\hat{h}\left(z\right), j\right) \left|h^T\left(z,j\right) - h^{S,i}\left(z,j\right)\right| p_g^{S,i}\left(z\right) dz$$

$$= L \sum_{i=1}^K \pi_i \sum_{j=1}^C \int_{\mathcal{Z}} \left|h^T\left(z,j\right) - h^{S,i}\left(z,j\right)\right| p_g^{S,i}\left(z\right) dz$$

$$\overset{(1)}{\leq} L \sum_{i=1}^K \pi_i \sum_{j=1}^C \int_{\mathcal{X}} \left|f^T\left(z,j\right) - f^{S,i}\left(z,j\right)\right| p^{S,i}\left(x\right) dx$$

$$\leq L \max_{i\in[K]} \mathbb{E}_{\mathbb{P}^{S,i}} \left[\left\|\Delta p^i\left(y|x\right)\right\|_1\right].$$

Here we note that $\overset{(1)}{\leq}$ results from (ii) in Proposition 1 with $c\left(\beta, i\right) = 1$ for any $\beta \in \mathcal{Y}_\Delta$ and $i \in [C]$. Furthermore, $\Delta p^i(y|x) := \left[\left|f^T(x,y) - f^{S,i}(x,y)\right|\right]_{y=1}^C$ is the absolute single point label shift on input space between the source domain $\mathbb{D}^{S,i}$ and the target domain $\mathbb{D}^T$.

With these two terms, we have the upper bound for hybrid domain as

$$\mathcal{L}\left(\hat{h}, \mathbb{D}_g^{hy}\right) \leq \sum_{i=1}^K \pi_i \mathcal{L}\left(\hat{f}, \mathbb{D}^{S,i}\right) + L \max_{i\in[K]} \mathbb{E}_{\mathbb{P}^{S,i}} \left[\left\|\Delta p^i(y|x)\right\|_1\right].$$

Next, we relate the loss on target $\mathbb{D}_g^T$ to hybrid domain $\mathbb{D}_g^{hy}$, which differs only at the feature marginals.

$$\left| \mathcal{L}\left(\hat{h}, \mathbb{D}_g^T\right) - \mathcal{L}\left(\hat{h}, \mathbb{D}_g^{hy}\right) \right| = \left| \int \ell\left(\hat{h}(z), h^T(z)\right) \left(p_g^T(z) - p_g^\pi(z)\right) dz \right|$$

$$\leq \int L \left| p_g^T(z) - p_g^\pi(z) \right| dz$$

$$\leq L \int \left| \sqrt{p_g^T(z)} + \sqrt{p_g^\pi(z)} \right| \left| \sqrt{p_g^T(z)} - \sqrt{p_g^\pi(z)} \right| dz$$

$$\leq L \left[ \int \left( \sqrt{p_g^T(z)} + \sqrt{p_g^\pi(z)} \right)^2 dz \right]^{1/2} \left[ \int \left( \sqrt{p_g^T(z)} - \sqrt{p_g^\pi(z)} \right)^2 dz \right]^{1/2}$$

$$\leq \frac{L}{\sqrt{2}} \left[ \int \left( p_g^T(z) + p_g^\pi(z) + 2\sqrt{p_g^T(z)p_g^\pi(z)} \right) dz \right]^{1/2}$$

$$\times \left[ 2 \int \left( \sqrt{p_g^T(z)} - \sqrt{p_g^\pi(z)} \right)^2 dz \right]^{1/2}$$

$$\leq \frac{L}{\sqrt{2}} \left[ 2 + 2 \left( \int p_g^T(z)dz \int p_g^\pi(z)dz \right)^{1/2} \right]^{1/2} d_{1/2}\left(\mathbb{P}_g^T, \mathbb{P}_g^\pi\right)$$

$$\leq L\sqrt{2} d_{1/2}\left(\mathbb{P}_g^T, \mathbb{P}_g^\pi\right).$$

In the above proof, we repeatedly invoke Cauchy-Schwartz inequality $\left| \int f(z)g(z)dz \right|^2 \leq \int |f(z)|^2 \, dz \int |g(z)|^2 \, dz$. Moreover, for the sake of completeness, we reintroduce the definition of the square root Hellinger distance

$$d_{1/2}\left(\mathbb{P}_g^T, \mathbb{P}_g^\pi\right) = \left[ 2 \int \left( \sqrt{p_g^T(z)} - \sqrt{p_g^\pi(z)} \right)^2 dz \right]^{1/2}.$$

To this end, we obtain the upper bound for target loss related to loss on souce mixture, a label shift term on input space, and a data shift term between target domain and source mixture on feature space.

$$\mathcal{L}\left(\hat{h}, \mathbb{D}_g^T\right) \leq \sum_{i=1}^{K} \pi_i \mathcal{L}\left(\hat{f}, \mathbb{D}^{S,i}\right) + L \max_{i \in [K]} \mathbb{E}_{\mathbb{P}^{S,i}} \left[ \|\Delta p^i(y|x)\|_1 \right] + L\sqrt{2} d_{1/2}\left(\mathbb{P}_g^T, \mathbb{P}_g^\pi\right). \quad (2)$$

Finally, using the fact that loss on feature space equal loss on input space (Corollary 2), we have

$$\mathcal{L}\left(\hat{f}, \mathbb{D}^T\right) \leq \sum_{i=1}^{K} \pi_i \mathcal{L}\left(\hat{f}, \mathbb{D}^{S,i}\right) + L \max_{i \in [K]} \mathbb{E}_{\mathbb{P}^{S,i}} \left[ \|\Delta p^i(y|x)\|_1 \right] + L\sqrt{2} d_{1/2}\left(\mathbb{P}_g^T, \mathbb{P}_g^\pi\right).$$

That concludes our proof. $\qquad \square$

This bound is novel since it relates loss on input space and data shift on feature space. This allows us to further investigate how source-source compression and source-target compression affect learning. First, we prove a lemma showing decomposition of data shift between target domain and source mixture $d_{1/2}\left(\mathbb{P}_g^T, \mathbb{P}_g^\pi\right)$ to a sum of data shifts between target domain and source domains.

**Lemma 4.** *Given a source mixture and a target domain, we have the following*

$$d_{1/2}\left(\mathbb{P}_g^T, \mathbb{P}_g^\pi\right) \leq \sum_{j=1}^{K} \sqrt{\pi_j} d_{1/2}\left(\mathbb{P}_g^T, \mathbb{P}_g^{S,j}\right)$$

*Proof.* Firstly, we observe that

$$d_{1/2}\left(\mathbb{P}_g^T, \mathbb{P}_g^\pi\right) = \left[2\int\left(\sqrt{p_g^T(z)} - \sqrt{p_g^\pi(z)}\right)^2 dz\right]^{1/2}$$

$$= \left[2\int\left(p_g^T(z) + p_g^\pi(z) - 2\sqrt{p_g^T(z)p_g^\pi(z)}\right)dz\right]^{1/2}.$$

Secondly, we use Cauchy-Schwartz inequality to obtain

$$p_g^T(z)p_g^\pi(z) = \left(\sum_{j=1}^K \pi_j p_g^T(z)\right)\left(\sum_{j=1}^K \pi_j p_g^{S,j}(z)\right)$$

$$\geq \left(\sum_{j=1}^K \pi_j\sqrt{p_g^T(z)p_g^{S,j}(z)}\right)^2.$$

Therefore, we arrive at

$$d_{1/2}\left(\mathbb{P}_g^T, \mathbb{P}_g^\pi\right) \leq \left[2\int\left(\sum_{j=1}^K \pi_j p_g^T(z) + \sum_{j=1}^K \pi_j p_g^{S,j}(z) - 2\sum_{j=1}^K \pi_j\sqrt{p_g^T(z)p_g^{S,j}(z)}\right)dz\right]^{1/2}$$

$$= \left[\sum_{j=1}^K \pi_j 2\int\left(p_g^T(z) + p_g^{S,j}(z) - 2\sqrt{p_g^T(z)p_g^{S,j}(z)}\right)dz\right]^{1/2}$$

$$\leq \sum_{j=1}^K \left[\pi_j 2\int\left(p_g^T(z) + p_g^{S,j}(z) - 2\sqrt{p_g^T(z)p_g^{S,j}(z)}\right)dz\right]^{1/2}$$

$$= \sum_{j=1}^K \sqrt{\pi_j}\, d_{1/2}\left(\mathbb{P}_g^T, \mathbb{P}_g^{S,j}\right).$$

$\square$

Now we are ready to prove the bound which motivate compressed DI representation.

**Theorem 5.** (*Theorem 3 in the main paper*) *Consider mixture of source domains* $\mathbb{D}^\pi = \sum_{i=1}^K \pi_i \mathbb{D}^{S,i}$ *and target domain* $\mathbb{D}^T$. *Let* $\ell$ *be any loss function upper-bounded by a positive constant* $L$. *For any hypothesis* $\hat{f}: \mathcal{X} \mapsto \mathcal{Y}_\Delta$ *where* $\hat{f} = \hat{h} \circ g$ *with* $g: \mathcal{X} \mapsto \mathcal{Z}$ *and* $\hat{h}: \mathcal{Z} \mapsto \mathcal{Y}_\Delta$, *the target loss on input space is upper bounded*

$$\mathcal{L}\left(\hat{f}, \mathbb{D}^T\right) \leq \sum_{i=1}^K \pi_i \mathcal{L}\left(\hat{f}, \mathbb{D}^{S,i}\right) + L\max_{i\in[K]}\mathbb{E}_{\mathbb{P}^{S,i}}\left[\|\Delta p^i(y|x)\|_1\right]$$

$$+ \sum_{i=1}^K\sum_{j=1}^K \frac{L\sqrt{2\pi_j}}{K}\left(d_{1/2}\left(\mathbb{P}_g^T, \mathbb{P}_g^{S,i}\right) + d_{1/2}\left(\mathbb{P}_g^{S,i}, \mathbb{P}_g^{S,j}\right)\right)$$

(3)

*Proof.* In the previous Theorem 3, the upper bound for target loss is

$$\mathcal{L}\left(\hat{f}, \mathbb{D}^T\right) \leq \sum_{i=1}^K \pi_i \mathcal{L}\left(\hat{f}, \mathbb{D}^{S,i}\right) + L\max_{i\in[K]}\mathbb{E}_{\mathbb{P}^{S,i}}\left[\|\Delta p^i(y|x)\|_1\right] + L\sqrt{2}d_{1/2}\left(\mathbb{P}_g^T, \mathbb{P}_g^\pi\right).$$

Using Lemma 4, we have

$$d_{1/2}\left(\mathbb{P}_g^T, \mathbb{P}_g^\pi\right) \leq \sum_{j=1}^K \sqrt{\pi_j} d_{1/2}\left(\mathbb{P}_g^T, \mathbb{P}_g^{S,j}\right)$$

Next, we use the triangle inequality for square root Hellinger distance

$$d_{1/2}\left(\mathbb{P}_g^T, \mathbb{P}_g^\pi\right) \leq \sum_{j=1}^K \sqrt{\pi_j} d_{1/2}\left(\mathbb{P}_g^T, \mathbb{P}_g^{S,j}\right)$$

$$\leq \sum_{j=1}^K \sqrt{\pi_j}\left(d_{1/2}\left(\mathbb{P}_g^T, \mathbb{P}_g^{S,i}\right) + d_{1/2}\left(\mathbb{P}_g^{S,i}, \mathbb{P}_g^{S,j}\right)\right)$$

Therefore, by average over all $\mathbb{P}_g^T, \mathbb{P}_g^{S,i}$ pairs,

$$d_{1/2}\left(\mathbb{P}_g^T, \mathbb{P}_g^\pi\right) = \sum_{i=1}^K \frac{1}{K} d_{1/2}\left(\mathbb{P}_g^T, \mathbb{P}_g^\pi\right) \leq \sum_{i=1}^K \sum_{j=1}^K \frac{\sqrt{\pi_j}}{K}\left(d_{1/2}\left(\mathbb{P}_g^T, \mathbb{P}_g^{S,i}\right) + d_{1/2}\left(\mathbb{P}_g^{S,i}, \mathbb{P}_g^{S,j}\right)\right)$$

We obtain the conclusion of our proof

$$\mathcal{L}\left(\hat{f}, \mathbb{D}^T\right) \leq \sum_{i=1}^K \pi_i \mathcal{L}\left(\hat{f}, \mathbb{D}^{S,i}\right) + L \max_{i \in [K]} \mathbb{E}_{\mathbb{P}^{S,i}}\left[\|\Delta p^i(y|x)\|_1\right]$$

$$+ \sum_{i=1}^K \sum_{j=1}^K \frac{L\sqrt{2\pi_j}}{K}\left(d_{1/2}\left(\mathbb{P}_g^T, \mathbb{P}_g^{S,i}\right) + d_{1/2}\left(\mathbb{P}_g^{S,i}, \mathbb{P}_g^{S,j}\right)\right)$$

$\square$

## 2 Appendix B: DI Representation's Characteristics

### 2.1 General Domain-Invariant Representations

In the main paper, we defined general DI representation via minimization of source loss $\min_{g \in \mathcal{G}} \min_{\hat{h} \in \mathcal{H}} \sum_{i=1}^K \pi_i \mathcal{L}\left(\hat{h}, h^{S,i}, \mathbb{P}_g^{S,i}\right)$. We then proposed to view the optimization problem $\min_{\hat{h} \in \mathcal{H}} \sum_{i=1}^K \pi_i \mathcal{L}\left(\hat{h}, h^{S,i}, \mathbb{P}_g^{S,i}\right)$ as calculating a type of divergence, i.e., hypothesis-aware divergence. To understand the connection between the two, we first consider the classification problem where samples are drawn from a mixture $z \sim \mathbb{Q}^\alpha = \sum_{i=1}^C \alpha_i \mathbb{Q}_i$, with $\mathbb{Q}_i$ defined on $\mathcal{Z}$ and density being $q_i(z)$, and the task is to predict which distributions $\mathbb{Q}_1, ..., \mathbb{Q}_C$ the samples originate from, i.e., labels being $1, ..., C$. Here, the hypothesis class $\mathcal{H}$ is assumed to have infinite capacity, and the objective is to minimize $\min_{\hat{h} \in \mathcal{H}} \mathcal{L}_{\mathbb{Q}_{1:C}}^\alpha\left(\hat{h}\right) = \min_{\hat{h} \in \mathcal{H}} \sum_{i=1}^C \alpha_i \mathcal{L}\left(\hat{h}, \mathbb{Q}_i\right)$.

#### 2.1.1 Hypothesis-Aware Divergence

**Theorem 6.** (***Theorem 5 in the main paper***) *Assuming the hypothesis class $\mathcal{H}$ has infinite capacity, we define the hypothesis-aware divergence for multiple distributions as*

$$D^\alpha\left(\mathbb{Q}_1, ..., \mathbb{Q}_C\right) = -\min_{\hat{h} \in \mathcal{H}} \mathcal{L}_{\mathbb{Q}_{1:C}}^\alpha\left(\hat{h}\right) + \inf_{\beta \in \mathcal{Y}_\Delta}\left(\sum_{i=1}^C l\left(\beta, i\right)\alpha_i\right). \tag{4}$$

*This divergence is a proper divergence among $\mathbb{Q}_1, ..., \mathbb{Q}_C$ in the sense that $D^\alpha\left(\mathbb{Q}_1, ..., \mathbb{Q}_C\right) \geq 0$ for all $\mathbb{Q}_1, ..., \mathbb{Q}_C$ and $\alpha \in \mathcal{Y}_\Delta$, and $D^\alpha\left(\mathbb{Q}_1, ..., \mathbb{Q}_C\right) = 0$ if $\mathbb{Q}_1 = ... = \mathbb{Q}_C$.*

*Proof.* Data is sampled from the mixture $\mathbb{Q}^\alpha$ by firstly sampling domain index $i \sim Cat(\alpha)$, then sampling data $z \sim \mathbb{Q}_i$ and label with $i$. We examine the the total expected loss for any hypothesis $\hat{h} \in \mathcal{H}$, which is

$$\mathcal{L}^\alpha_{\mathbb{Q}_{1:C}}\left(\hat{h}\right) := \sum_{i=1}^{C} \alpha_i \mathcal{L}\left(\hat{h}, \mathbb{Q}_i\right)$$

$$= \sum_{i=1}^{C} \alpha_i \int l\left(\hat{h}(z), i\right) q_i(z) \, dz$$

We would like to minimize this loss, which leads to

$$
\begin{aligned}
\min_{\hat{h} \in \mathcal{H}} \mathcal{L}^\alpha_{\mathbb{Q}_{1:C}}\left(\hat{h}\right) &= \min_{\hat{h} \in \mathcal{H}} \sum_{i=1}^{C} \alpha_i \int l\left(\hat{h}(z), i\right) q_i(z) \, dz \\
&\overset{(1)}{=} \min_{\hat{h} \in \mathcal{H}} \int \left(\sum_{i=1}^{C} \alpha_i l\left(\hat{h}(z), i\right) \frac{q_i(z)}{q^\alpha(z)}\right) q^\alpha(z) \, dz \\
&\overset{(2)}{=} \int \min_{\hat{h} \in \mathcal{H}} \left(\sum_{i=1}^{C} \alpha_i l\left(\hat{h}(z), i\right) \frac{q_i(z)}{q^\alpha(z)}\right) q^\alpha(z) \, dz \\
&= \int \min_{\beta \in \mathcal{Y}_\Delta} \left(\sum_{i=1}^{C} \alpha_i l(\beta, i) \frac{q_i(z)}{q^\alpha(z)}\right) q^\alpha(z) \, dz \\
&\overset{(3)}{\leq} \min_{\beta \in \mathcal{Y}_\Delta} \left(\int \sum_{i=1}^{C} \alpha_i l(\beta, i) q_i(z) \, dz\right) \\
&= \min_{\beta \in \mathcal{Y}_\Delta} \left(\sum_{i=1}^{C} \alpha_i l(\beta, i)\right).
\end{aligned}
\tag{5}
$$

For (1), $q^\alpha(z) = \sum_{i=1}^{C} \alpha_i q_i(z)$ is introduced as the density of the mixture, whereas for (2), we use the fact that $\mathcal{H}$ has infinite capacity, leading to the equality $\min_{\hat{h} \in \mathcal{H}} \int f\left(\hat{h}(x)\right) q(x) \, dx = \int \min_{\hat{h} \in \mathcal{H}} f\left(\hat{h}(x)\right) q(x) \, dx$. Moreover, for (3), the property of concave function $\phi(t) = \min_{\beta \in \mathcal{Y}_\Delta}\left(\sum_{i=1}^{C} \alpha_i l(\beta, i) t_i\right)$ with $t \in \mathcal{Y}_\Delta$ is invoked, i.e., $\mathbb{E}_{z \sim Q}\left[\phi(t(z))\right] \leq \phi\left(\mathbb{E}_{z \sim Q}\left[t(z)\right]\right)$.

This hints us to define a non-zero divergence $D^\alpha$ between multiple distributions $\mathbb{Q}_1, ..., \mathbb{Q}_C$ as

$$
\begin{aligned}
D^\alpha\left(\mathbb{Q}_1, ..., \mathbb{Q}_C\right) &= -\min_{\hat{h} \in \mathcal{H}} \mathcal{L}^\alpha_{\mathbb{Q}_{1:C}}\left(\hat{h}\right) + \inf_{\beta \in \mathcal{Y}_\Delta}\left(\sum_{i=1}^{C} l(\beta, i) \alpha_i\right), \\
&= \int -\phi\left(\left[\frac{q_i(z)}{q^\alpha(z)}\right]_{i=1}^{C}\right) q^\alpha(z) \, dz + \inf_{\beta \in \mathcal{Y}_\Delta}\left(\sum_{i=1}^{C} l(\beta, i) \alpha_i\right)
\end{aligned}
$$

which is a proper f-divergence, since $-\phi(t)$ is a convex function, and $\inf_{\beta \in \mathcal{Y}_\Delta}\left(\sum_{i=1}^{C} l(\beta, i) \alpha_i\right)$ is just a constant. Moreover, $D^\alpha\left(\mathbb{Q}_1, ..., \mathbb{Q}_C\right) \geq 0$ for all $\mathbb{Q}_1, ..., \mathbb{Q}_C$ and $\alpha \in \mathcal{Y}_\Delta$ due to the previous inequality 5. The equality happens if there is some $\beta_0 \in \mathcal{Y}_\Delta$ such that, for all $z \in \mathcal{Z}$

$$\beta_0 = \underset{\beta \in \mathcal{Y}_\Delta}{\operatorname{argmin}} \sum_{i=1}^{C} \alpha_i l(\beta, i) \frac{q_i(z)}{q^\alpha(z)}.$$

This means $\frac{q_i(z)}{q^\alpha(z)} = A_i, \forall i \in [C]$, where $A_i$ is a constant dependent on index $i$. However, this leads to

$$\int q_i(z)\, dz = A_i \int q^\alpha(z)\, dz$$
$$1 = A_i$$

i.e., $q_i(z) = q^\alpha(z), \forall i \in [C]$. In other words, the equality happens when all distributions are the same $\mathbb{Q}_1 = ... = \mathbb{Q}_C$. □

### 2.1.2 General Domain-Invariant Representations

For a fixed feature map $g$, the induced representation distributions of source domains are $\mathbb{P}_g^{S,i}$. We then find the optimal hypothesis $\hat{h}_g^*$ on the induced representation distributions $\mathbb{P}_g^{S,i}$ by minimizing the loss

$$\min_{\hat{h} \in \mathcal{H}} \sum_{i=1}^{K} \pi_i \mathcal{L}\left(\hat{h}, h^{S,i}, \mathbb{P}_g^{S,i}\right) = \min_{\hat{h} \in \mathcal{H}} \sum_{i=1}^{K} \pi_i \mathcal{L}\left(\hat{h}, \mathbb{D}_g^{S,i}\right). \tag{6}$$

The general domain-invariant feature map $g^*$ is defined as the one that offers the minimal optimal loss as

$$g^* = \arg\min_{g \in \mathcal{G}} \min_{\hat{h} \in \mathcal{H}} \sum_{i=1}^{K} \pi_i \mathcal{L}\left(\hat{h}, h^{S,i}, \mathbb{P}_g^i\right) = \operatorname{argmin}_{g \in \mathcal{G}} \min_{\hat{h} \in \mathcal{H}} \sum_{i=1}^{K} \pi_i \mathcal{L}\left(\hat{h}, \mathbb{D}_g^{S,i}\right). \tag{7}$$

We denote $\mathbb{P}_g^{s,i,c}$ as the class $c$ conditional distribution of the source domain $i$ on the latent space and $p_g^{s,i,c}$ as its density function. The induced representation distribution $\mathbb{P}_g^{S,i}$ of source domain $i$ is a mixture of $\mathbb{P}_g^{s,i,c}$ as $\mathbb{P}_g^{S,i} = \sum_{c=1}^{C} \gamma_{i,c} \mathbb{P}_g^{s,i,c}$, where $\gamma_{i,c} = \mathbb{P}^{s,i}(y = c)$.

We further define $\mathbb{Q}_g^{s,c} := \sum_{i=1}^{K} \frac{\pi_i \gamma_{i,c}}{\alpha_c} \mathbb{P}_g^{s,i,c}$ where $\alpha_c = \sum_{j=1}^{K} \pi_j \gamma_{j,c}$. Obviously, we can interpret $\mathbb{Q}_g^{s,c}$ as the mixture of the class $c$ conditional distributions of the source domains on the latent space. The objective function in Eq. (6) can be viewed as training the optimal hypothesis $\hat{h} \in \mathcal{H}$ to distinguish the samples from $\mathbb{Q}_g^{s,c}, c \in [C]$ for a given feature map $g$. Therefore, by linking to the multi-divergence concept developed in Theorem 6, we achieve the following theorem.

**Theorem 7.** *(**Theorem 6 in the main paper**) Assume that $\mathcal{H}$ has infinite capacity, we have the following statements.*

*1. $D^\alpha\left(\mathbb{Q}_g^{s,1}, ..., \mathbb{Q}_g^{s,C}\right) = -\min_{\hat{h} \in \mathcal{H}} \sum_{i=1}^{K} \pi_i \mathcal{L}\left(\hat{h}, h^{S,i}, \mathbb{P}_g^{S,i}\right) + const$, where $\alpha = [\alpha_c]_{c \in [C]}$ is defined as above.*

*2. Finding the general domain-invariant feature map $g^*$ via the OP in (7) is equivalent to solving*

$$g^* = \operatorname*{argmax}_{g \in \mathcal{G}} D^\alpha\left(\mathbb{Q}_g^{s,1}, ..., \mathbb{Q}_g^{s,C}\right). \tag{8}$$

*Proof.* We investigate the loss on mixture

$$\sum_{i=1}^{K} \pi_i \mathcal{L}\left(\hat{h}, h^{S,i}, \mathbb{P}_g^{S,i}\right) = \sum_{i=1}^{K} \pi_i \sum_{c=1}^{C} \gamma_{i,c} \mathcal{L}\left(\hat{h}, c, \mathbb{P}_g^{S,i,c}\right)$$
$$= \sum_{c=1}^{C} \alpha_c \mathcal{L}\left(\hat{h}, c, \sum_{i=1}^{K} \frac{\pi_i \gamma_{i,c}}{\alpha_c} \mathbb{P}_g^{S,i,c}\right)$$
$$= \sum_{c=1}^{C} \alpha_c \mathcal{L}\left(\hat{h}, c, \mathbb{Q}_g^{S,c}\right)$$

Therefore, the loss on mixture is actually a loss on joint class-conditional distributions $\mathbb{Q}_g^{S,c} = \sum_{i=1}^{K} \frac{\pi_i \gamma_{i,c}}{\alpha_c} \mathbb{P}_g^{S,i,c}$. Using result from Theorem 6, we can define a divergence between these class-conditionals

$$D^{\alpha}\left(\mathbb{Q}_g^{S,1}, ..., \mathbb{Q}_g^{S,C}\right) = -\min_{\hat{h} \in \mathcal{H}} \sum_{c=1}^{C} \alpha_c \mathcal{L}\left(\hat{h}, \mathbb{Q}_g^{S,c}\right) + \min_{\beta \in \mathcal{Y}_\Delta}\left(\sum_{c=1}^{C} \ell\left(\beta, i\right) \alpha_c\right)$$

$$= -\sum_{i=1}^{K} \pi_i \mathcal{L}\left(\hat{h}, h^{S,i}, \mathbb{P}_g^{S,i}\right) + \text{const}$$

$\square$

## 2.2 Compressed Domain-Invariant Representations

**Theorem 8.** *(**Theorem 7 in the main paper**) For any confident level $\delta \in [0,1]$ over the choice of $S$, the estimation of loss is in the $\epsilon$-range of the true loss*

$$Pr\left(\left|\mathcal{L}\left(\hat{h}, S\right) - \mathcal{L}\left(\hat{h}, \mathbb{D}_g^\pi\right)\right| \leq \epsilon\right) \geq 1 - \delta,$$

*where $\epsilon = \epsilon\left(\delta\right) = \left(\frac{A}{\delta}\right)^{1/2}$ is a function of $\delta$ for which $A$ is proportional to*

$$\frac{1}{N}\left(\sum_{i=1}^{K}\sum_{j=1}^{K} \frac{\sqrt{\pi_i}}{K} \mathcal{L}\left(\hat{f}, \mathbb{D}^{S,j}\right) + L\sum_{i=1}^{K} \sqrt{\pi_i} \max_{k \in [K]} \mathbb{E}_{\mathbb{P}^{S,k}}\left[\left\|\Delta p^{k,i}\left(y|x\right)\right\|_1\right] + \frac{L}{K}\sum_{i=1}^{K}\sum_{j=1}^{K} \sqrt{2\pi_i} \, d_{1/2}\left(\mathbb{P}_g^{S,i}, \mathbb{P}_g^{S,j}\right)\right)^2.$$

*Proof.* Let $S$ be a sample of $N$ data points $(z, y) \sim \mathbb{D}_g^\pi$ sampled from the mixture domain, i.e., i.e., $i \sim Cat(\pi)$, $z \sim \mathbb{P}^{S,i}$, and labeling with corresponding $y \sim Cat\left(\hat{h}^{S,i}(z)\right)$. The loss of a hypothesis $h$ on a sample $(z, y) = \left(z, \hat{h}^{S,i}(z)\right)$ for some domain index $i$ is $\ell\left(\hat{h}(z), \hat{h}^{S,i}(z)\right)$. To avoid crowded notation, we denote this loss as $\ell^i(z) := \ell\left(\hat{h}(z), \hat{h}^{S,i}(z)\right)$.

Let $N = \sum_{i=1}^{K} N_i$, where each $N_i$ is the number of sample drawn from domain $i$. The estimation of loss on a particular domain $i$ is

$$\mathcal{L}\left(\hat{h}, S^i\right) = \sum_{j=1}^{N_i} \frac{1}{N_i} \ell^i(z_j).$$

This estimation is unbiased estimation, i.e., $\mathbb{E}_{S^i \sim \left(\mathbb{D}_g^{S,i}\right)^{N_i}}\left[\mathcal{L}\left(\hat{h}, S^i\right)\right] = \mathbb{E}_{z \sim \mathbb{P}_g^{S,i}}\left[\ell^i(z)\right] = \mathcal{L}\left(\hat{h}, \mathbb{D}_g^{S,i}\right)$. Furthermore, loss estimation on source mixture is

$$\mathcal{L}(h, S) = \sum_{i=1}^{K}\sum_{j=1}^{N_i} \frac{1}{N} \ell^i(z_j)$$

This estimation is also an unbiased estimation

$$\mathbb{E}_{S \sim \left(\mathbb{D}_g^\pi\right)^N}\left[\mathcal{L}\left(\hat{h}, S\right)\right] = \mathbb{E}_{\{N_i\}}\left[\mathbb{E}_{S^i}\left[\mathcal{L}\left(\hat{h}, S\right)\right]\right]$$

$$= \sum_{i=1}^{K} \pi_i \mathcal{L}\left(\hat{h}, \mathbb{D}_g^{S,i}\right) = \mathcal{L}\left(\hat{h}, \mathbb{D}_g^\pi\right).$$

Therefore, we can bound the concentration of $\mathcal{L}\left(\hat{h}, S\right)$ around its mean value $\mathcal{L}\left(\hat{h}, \mathbb{D}_g^\pi\right)$ using Chebyshev's inequality

$$Pr\left(\left|\mathcal{L}\left(\hat{h}, S\right) - \mathcal{L}\left(\hat{h}, \mathbb{D}_g^\pi\right)\right| \leq \epsilon\right) \geq 1 - \frac{\text{Var}_{S \sim \left(\mathbb{D}_g^\pi\right)^N}\left[\mathcal{L}\left(\hat{h}, S\right)\right]}{\epsilon^2}$$

which is equivalent to

$$\Pr\left(\left|\mathcal{L}\left(\hat{h},S\right)-\mathcal{L}\left(\hat{h},\mathbb{D}_g^\pi\right)\right|\leq\sqrt{\frac{\mathrm{Var}_{S\sim\left(\mathbb{D}_g^\pi\right)^N}\left[\mathcal{L}\left(\hat{h},S\right)\right]}{\delta}}\right)\geq1-\delta$$

The variance of $\mathcal{L}\left(\hat{h},S\right)$ is

$$
\begin{aligned}
\mathrm{Var}_{S\sim\left(\mathbb{D}_g^\pi\right)^N}\left[\mathcal{L}\left(\hat{h},S\right)\right] &\overset{(1)}{=} \frac{1}{N}\mathrm{Var}_{(z,y)\sim\mathbb{D}_g^\pi}\left[\ell\left(\hat{h}(z),y\right)\right] \\
&\overset{(2)}{=} \frac{1}{N}\sum_{i=1}^K \pi_i\left(\mathrm{Var}_{z\sim\mathbb{P}_g^{S,i}}\left[\ell^i\left(z\right)\right]+\mathbb{E}_{z\sim\mathbb{P}_g^{S,i}}\left[\ell^i\left(z\right)\right]^2\right)-\left(\mathbb{E}_{(z,y)\sim\mathbb{D}_g^\pi}\left[\ell\left(\hat{h}(z),y\right)\right]\right)^2 \\
&\leq \frac{1}{N}\sum_{i=1}^K \pi_i\left(\mathrm{Var}_{z\sim\mathbb{P}_g^{S,i}}\left[\ell^i\left(z\right)\right]+\mathcal{L}\left(\hat{h},\mathbb{D}_g^{S,i}\right)^2\right) \\
&\leq \frac{1}{N}\sum_{i=1}^K \pi_i\mathrm{Var}_{z\sim\mathbb{P}_g^{S,i}}\left[\ell^i\left(z\right)\right]+\frac{1}{N}\left(\sum_{i=1}^K\sqrt{\pi_i}\mathcal{L}\left(\hat{h},\mathbb{D}_g^{S,i}\right)\right)^2
\end{aligned}
$$
(9)

$\overset{(1)}{=}$ is true since $\mathcal{L}\left(h,S\right)$ is the sum of $N$ i.i.d. random variable $\ell\left(h(z),y\right)$ with $(z,y)$ sampled from the same distribution $\mathbb{D}_g^\pi$. In (2), the variance of w.r.t. a distribution mixture is related to mean and variance of constituting distribution, i.e., $\mathrm{Var}_{\sum_i \pi_i\mathbb{P}_i}\left[X\right]=\sum_i \pi_i\left(\mathrm{Var}_{\mathbb{P}_i}\left[X\right]+\mathbb{E}_{\mathbb{P}_i}\left[X\right]^2\right)-\mathbb{E}_{\sum_i \pi_i\mathbb{P}_i}\left[X\right]^2$.

We reuse the result of 2 in Theorem 3, substituting $\mathbb{D}_g^T\equiv\mathbb{D}_g^{S,i},\mathbb{D}_g^\pi\equiv\mathbb{D}_g^{S,j}$ to obtain

$$
\begin{aligned}
\mathcal{L}\left(\hat{h},\mathbb{D}_g^{S,i}\right) &\leq \mathcal{L}\left(\hat{f},\mathbb{D}^{S,j}\right)+L\mathbb{E}_{\mathbb{P}^{S,i}}\left[\|\Delta p^{i,j}(y|x)\|_1\right]+L\sqrt{2}\,d_{1/2}\left(\mathbb{P}_g^{S,i},\mathbb{P}_g^{S,j}\right) \\
&\leq \frac{1}{K}\sum_{j=1}^K\left(\mathcal{L}\left(\hat{f},\mathbb{D}^{S,j}\right)+L\max_{k\in[K]}\mathbb{E}_{\mathbb{P}^{S,k}}\left[\|\Delta p^{k,i}\left(y|x\right)\|_1\right]+L\sqrt{2}\,d_{1/2}\left(\mathbb{P}_g^{S,i},\mathbb{P}_g^{S,j}\right)\right).
\end{aligned}
$$

Therefore, the right hand side of 9 is upper by $A$, where $A$ is

$$
\begin{aligned}
A = &\frac{1}{N}\sum_{i=1}^K \pi_i\mathrm{Var}_{z\sim\mathbb{P}_g^{S,i}}\left[\ell^i\left(z\right)\right] \\
&+\frac{1}{N}\left(\sum_{i=1}^K\sum_{j=1}^K\frac{\sqrt{\pi_i}}{K}\mathcal{L}\left(\hat{f},\mathbb{D}^{S,j}\right)+L\sum_{i=1}^K\sqrt{\pi_i}\max_{k\in[K]}\mathbb{E}_{\mathbb{P}^{S,k}}\left[\left\|\Delta p^{k,i}\left(y|x\right)\right\|_1\right]+\frac{L}{K}\sum_{i=1}^K\sum_{j=1}^K\sqrt{2\pi_i}\,d_{1/2}\left(\mathbb{P}_g^{S,i},\mathbb{P}_g^{S,j}\right)\right)^2.
\end{aligned}
$$

This concludes our proof, where the concentration inequality is

$$\Pr\left(\left|\mathcal{L}\left(\hat{h},S\right)-\mathcal{L}\left(\hat{h},\mathbb{D}_g^\pi\right)\right|\leq\sqrt{\frac{A}{\delta}}\right)\geq1-\delta\quad.$$

$\square$

# 3    Appendix C: Trade-Off in Learning DI Representations

**Lemma 9.** *Given a labeling function $f:\mathcal{X}\to\mathcal{Y}_\Delta$ and a hypothesis $\hat{f}:\mathcal{X}\to\mathcal{Y}_\Delta$, let denote $\mathbb{P}_{\mathcal{Y}}^f$ and $\mathbb{P}_{\mathcal{Y}}^{\hat{f}}$ as two label marginal distributions induced by $f$ and $\hat{f}$ on the data distribution $\mathbb{P}$. Particularly,*

to sample $y \sim \mathbb{P}_{\mathcal{Y}}^f$, we first sample $x \sim \mathbb{P}$ (i.e., $\mathbb{P}$ is the data distribution with the density function $p$) and then sample $y \sim Cat\left(f\left(x\right)\right)$, while similar to sample $y \sim \mathbb{P}_{\mathcal{Y}}^{\hat{f}}$. We then have

$$d_{1/2}\left(\mathbb{P}_{\mathcal{Y}}^f, \mathbb{P}_{\mathcal{Y}}^{\hat{f}}\right) \leq \mathcal{L}\left(\hat{f}, f, \mathbb{P}\right)^{1/2},$$

where the loss $\mathcal{L}$ is defined based on the Hellinger loss $\ell\left(\hat{f}\left(x\right), f\left(x\right)\right) = D_{1/2}\left(\hat{f}\left(x\right), f\left(x\right)\right) = 2\sum_{i=1}^{C}\left[\sqrt{\hat{f}\left(x,i\right)} - \sqrt{f\left(x,i\right)}\right]^2$.

*Proof.* We have

$$
\begin{aligned}
D_{1/2}\left(\mathbb{P}_{\mathcal{Y}}^f, \mathbb{P}_{\mathcal{Y}}^{\hat{f}}\right) &= 2\sum_{i=1}^{C}\left(\sqrt{p^f\left(y\right)} - \sqrt{p^{\hat{f}}\left(y\right)}\right)^2\\
&=2\sum_{i=1}^{C}\left(\sqrt{\int p^f\left(y=i \mid x\right)p(x)dx} - \sqrt{\int p^{\hat{f}}\left(y=i \mid x\right)p(x)dx}\right)^2\\
&=2\sum_{i=1}^{C}\left(\sqrt{\int f\left(x,i\right)p(x)dx} - \sqrt{\int \hat{f}\left(x,i\right)p(x)dx}\right)^2\\
&=2\sum_{i=1}^{C}\left[\int f\left(x,i\right)p(x)dx + \int \hat{f}\left(x,i\right)p(x)dx - 2\sqrt{\int f\left(x,i\right)p(x)dx}\sqrt{\int \hat{f}\left(x,i\right)p(x)dx}\right]\\
&\overset{(1)}{\leq} 2\sum_{i=1}^{C}\left[\int f\left(x,i\right)p(x)dx + \int \hat{f}\left(x,i\right)p(x)dx - 2\sqrt{\int f\left(x,i\right)\hat{f}\left(x,i\right)p(x)dx}\right]\\
&=2\sum_{i=1}^{C}\int\left[\sqrt{f\left(x,i\right)} - \sqrt{\hat{f}\left(x,i\right)}\right]^2 p(x)dx = \int 2\sum_{i=1}^{C}\left[\sqrt{f\left(x,i\right)} - \sqrt{\hat{f}\left(x,i\right)}\right]^2 p(x)dx\\
&=\int D_{1/2}\left(\hat{f}\left(x\right), f\left(x\right)\right)p(x)dx = \mathcal{L}\left(\hat{f}, f, \mathbb{P}\right),
\end{aligned}
$$

where we note that in the derivation in $\overset{(1)}{\leq}$, we use Cauchy-Schwarz inequality: $\int \hat{f}\left(x,i\right)p\left(x\right)dx \int f\left(x,i\right)p\left(x\right)dx \geq \left(\int \sqrt{\hat{f}\left(x,i\right)f\left(x,i\right)}p\left(x\right)dx\right)^2$.

Therefore, we reach the conclusion as

$$d_{1/2}\left(\mathbb{P}_{\mathcal{Y}}^f, \mathbb{P}_{\mathcal{Y}}^{\hat{f}}\right) \leq \mathcal{L}\left(\hat{f}, f, \mathbb{P}\right)^{1/2}.$$

$\square$

**Lemma 10.** *Consider the hypothesis $\hat{f} = \hat{h} \circ g$. We have the following inequalities w.r.t. the source and target domains:*

*(i)* $d_{1/2}\left(\hat{\mathbb{P}}_{\mathcal{Y}}^T, \mathbb{P}_{\mathcal{Y}}^T\right) \leq \mathcal{L}\left(\hat{h} \circ g, f^T, \mathbb{P}^T\right)^{1/2}$, *where $\mathbb{P}_{\mathcal{Y}}^T$ is the label marginal distribution induced by $f^T$ on $\mathbb{P}^T$, while $\hat{\mathbb{P}}_{\mathcal{Y}}^T$ is the label marginal distribution induced by $\hat{f}$ on $\mathbb{P}^T$.*

*(ii)* $d_{1/2}\left(\mathbb{P}_{\mathcal{Y}}^\pi, \hat{\mathbb{P}}_{\mathcal{Y}}^\pi\right) \leq \left[\sum_{i=1}^{K}\pi_i\mathcal{L}\left(\hat{h} \circ g, f^{S,i}, \mathbb{P}^{S,i}\right)\right]^{1/2}$, *where $\mathbb{P}_{\mathcal{Y}}^\pi := \sum_{i=1}^{K}\pi_i\mathbb{P}_{\mathcal{Y}}^{S,i}$ with $\mathbb{P}_{\mathcal{Y}}^{S,i}$ to be induced by $f^{S,i}$ on $\mathbb{P}^{S,i}$ and $\hat{\mathbb{P}}_{\mathcal{Y}}^\pi := \sum_{i=1}^{K}\pi_i\hat{\mathbb{P}}_{\mathcal{Y}}^{S,i}$ with $\hat{\mathbb{P}}_{\mathcal{Y}}^{S,i}$ to be induced by $\hat{f}$ on $\mathbb{P}^{S,i}$ (i.e., equivalently, the label marginal distribution induced by $\hat{f}$ on $\mathbb{P}^\pi := \sum_{i=1}^{K}\pi_i\mathbb{P}^{S,i}$).*

*Proof.* (i) The proof of this part is obvious from Lemma 9 by considering $f^T$ as $f$ and $\mathbb{P}^T$ as $\mathbb{P}$.

(ii) By the convexity of $D_{1/2}$, which is a member of $f$-divergence family, we have

$$D_{1/2}\left(\mathbb{P}_{\mathcal{Y}}^{\pi}, \hat{\mathbb{P}}_{\mathcal{Y}}^{\pi}\right) = D_{1/2}\left(\sum_{i=1}^{K}\pi_i\mathbb{P}_{\mathcal{Y}}^{S,i}, \sum_{i=1}^{K}\pi_i\hat{\mathbb{P}}_{\mathcal{Y}}^{S,i}\right) \leq \sum_{i=1}^{K}\pi_i D_{1/2}\left(\mathbb{P}_{\mathcal{Y}}^{S,i}, \hat{\mathbb{P}}_{\mathcal{Y}}^{S,i}\right)$$

$$\overset{(1)}{\leq} \sum_{i=1}^{K}\pi_i\mathcal{L}\left(\hat{h}\circ g, f^{S,i}, \mathbb{P}^{S,i}\right),$$

where the derivation in $\overset{(1)}{\leq}$ is from Lemma 9. Therefore, we reach the conclusion as

$$d_{1/2}\left(\mathbb{P}_{\mathcal{Y}}^{\pi}, \hat{\mathbb{P}}_{\mathcal{Y}}^{\pi}\right) \leq \left[\sum_{i=1}^{K}\pi_i\mathcal{L}\left(\hat{h}\circ g, f^{S,i}, \mathbb{P}^{S,i}\right)\right]^{1/2}.$$

$\square$

**Theorem 11.** *(**Theorem 8 in the main paper**) Consider a feature extractor g and a hypothesis $\hat{h}$, the Hellinger distance between two label marginal distributions $\mathbb{P}_{\mathcal{Y}}^{\pi}$ and $\mathbb{P}_{\mathcal{Y}}^{T}$ can be upper-bounded as:*

*(i)* $d_{1/2}\left(\mathbb{P}_{\mathcal{Y}}^{\pi}, \mathbb{P}_{\mathcal{Y}}^{T}\right) \leq \left[\sum_{k=1}^{K}\pi_k\mathcal{L}\left(\hat{h}\circ g, f^{S,k}, \mathbb{P}^{S,k}\right)\right]^{1/2} + d_{1/2}\left(\mathbb{P}_{g}^{T}, \mathbb{P}_{g}^{\pi}\right) + \mathcal{L}\left(\hat{h}\circ g, f^{T}, \mathbb{P}^{T}\right)^{1/2}.$

*(ii)* $d_{1/2}\left(\mathbb{P}_{\mathcal{Y}}^{\pi}, \mathbb{P}_{\mathcal{Y}}^{T}\right) \leq \left[\sum_{i=1}^{K}\pi_i\mathcal{L}\left(\hat{h}\circ g, f^{S,i}, \mathbb{P}^{S,i}\right)\right]^{1/2} + \sum_{i=1}^{K}\sum_{j=1}^{K}\frac{\sqrt{\pi_j}}{K}d_{1/2}\left(\mathbb{P}_{g}^{S,i}, \mathbb{P}_{g}^{S,j}\right) +$
$\sum_{i=1}^{K}\sum_{j=1}^{K}\frac{\sqrt{\pi_j}}{K}d_{1/2}\left(\mathbb{P}_{g}^{T}, \mathbb{P}_{g}^{S,i}\right) + \mathcal{L}\left(\hat{h}\circ g, f^{T}, \mathbb{P}^{T}\right)^{1/2}.$

*Here we note that the general loss $\mathcal{L}$ is defined based on the Hellinger loss $\ell$ defined as $\ell\left(\hat{f}(x), f(x)\right) = D_{1/2}\left(\hat{f}(x), f(x)\right).$*

*Proof.* (i) We define $\mathbb{P}_{\mathcal{Y}}^{\pi}, \hat{\mathbb{P}}_{\mathcal{Y}}^{\pi}$ and $\mathbb{P}_{\mathcal{Y}}^{T}, \hat{\mathbb{P}}_{\mathcal{Y}}^{T}$ as in Lemma 10. Recap that to sample $y \sim \hat{\mathbb{P}}_{\mathcal{Y}}^{\pi}$, we sample $k \sim Cat(\pi), x \sim \mathbb{P}^{S,k}$ (i.e., $x \sim \mathbb{P}^{\pi} := \sum_{k=1}^{K}\pi_k\mathbb{P}^{S,k}$), compute $z = g(x)$ (i.e., $z \sim \mathbb{P}_g^{\pi}$), and $y \sim Cat\left(\hat{h}(z)\right)$, while similar to draw $y \sim \hat{\mathbb{P}}_{\mathcal{Y}}^{T}$.

Using triangle inequality for Hellinger distance, we have

$$d_{1/2}\left(\mathbb{P}_{\mathcal{Y}}^{\pi}, \mathbb{P}_{\mathcal{Y}}^{T}\right) \leq d_{1/2}\left(\mathbb{P}_{\mathcal{Y}}^{\pi}, \hat{\mathbb{P}}_{\mathcal{Y}}^{\pi}\right) + d_{1/2}\left(\hat{\mathbb{P}}_{\mathcal{Y}}^{\pi}, \hat{\mathbb{P}}_{\mathcal{Y}}^{T}\right) + d_{1/2}\left(\hat{\mathbb{P}}_{\mathcal{Y}}^{T}, \mathbb{P}_{\mathcal{Y}}^{T}\right).$$

Referring to Lemma 10, we achieve

$$d_{1/2}\left(\mathbb{P}_{\mathcal{Y}}^{\pi}, \mathbb{P}_{\mathcal{Y}}^{T}\right) \leq \left[\sum_{i=1}^{K}\pi_i\mathcal{L}\left(\hat{h}\circ g, f^{S,i}, \mathbb{P}^{S,i}\right)\right]^{1/2} + d_{1/2}\left(\mathbb{P}_{\hat{\mathcal{Y}}}^{\pi}, \mathbb{P}_{\hat{\mathcal{Y}}}^{T}\right) + \mathcal{L}\left(\hat{h}\circ g, f^{T}, \mathbb{P}^{T}\right)^{1/2}.$$

From the monotonicity of Hellinger distance, when applying to $\mathbb{P}_g^{T}$ and $\mathbb{P}_g^{\pi}$ with the same transition probability $p(y = i \mid z) = \hat{h}(z, i)$ for obtaining $\mathbb{P}_{\hat{\mathcal{Y}}}^{\pi}$ and $\mathbb{P}_{\hat{\mathcal{Y}}}^{T}$, we have

$$d_{1/2}\left(\mathbb{P}_{\hat{\mathcal{Y}}}^{\pi}, \mathbb{P}_{\hat{\mathcal{Y}}}^{T}\right) \leq d_{1/2}\left(\mathbb{P}_{g}^{\pi}, \mathbb{P}_{g}^{T}\right).$$

Finally, we reach the conclusion as

$$d_{1/2}\left(\mathbb{P}_{\mathcal{Y}}^{\pi}, \mathbb{P}_{\mathcal{Y}}^{T}\right) \leq \left[\sum_{i=1}^{K}\pi_i\mathcal{L}\left(\hat{h}\circ g, f^{S,i}, \mathbb{P}^{S,i}\right)\right]^{1/2} + d_{1/2}\left(\mathbb{P}_{g}^{\pi}, \mathbb{P}_{g}^{T}\right) + \mathcal{L}\left(\hat{h}\circ g, f^{T}, \mathbb{P}^{T}\right)^{1/2}.$$

(ii) From Lemma 4, we can decompose the data shift term and use triangle inequality again, hence arriving at

$$d_{1/2}\left(\mathbb{P}_{\mathcal{Y}}^{\pi},\mathbb{P}_{\mathcal{Y}}^{T}\right) \leq \left[\sum_{i=1}^{K}\pi_i\mathcal{L}\left(\hat{h}\circ g, f^{S,i}, \mathbb{P}^{S,i}\right)\right]^{1/2} + \sum_{j=1}^{K}\sqrt{\pi_j}d_{1/2}\left(\mathbb{P}_g^{T}, \mathbb{P}_g^{S,j}\right) + \mathcal{L}\left(\hat{h}\circ g, f^{T}, \mathbb{P}^{T}\right)^{1/2}$$

$$\leq \left[\sum_{i=1}^{K}\pi_i\mathcal{L}\left(\hat{h}\circ g, f^{S,i}, \mathbb{P}^{S,i}\right)\right]^{1/2} + \sum_{i=1}^{K}\sum_{j=1}^{K}\frac{\sqrt{\pi_j}}{K}\left(d_{1/2}\left(\mathbb{P}_g^{T}, \mathbb{P}_g^{S,i}\right) + d_{1/2}\left(\mathbb{P}_g^{S,i}, \mathbb{P}_g^{S,j}\right)\right)$$

$$+ \mathcal{L}\left(\hat{h}\circ g, f^{T}, \mathbb{P}^{T}\right)^{1/2}.$$

This concludes our proof. $\qquad\square$

The loss $\mathcal{L}$ in Theorem 11 defined based on the Hellinger loss $\ell$ defined as $\ell\left(\hat{f}\left(x\right), f(x)\right) = D_{1/2}\left(\hat{f}\left(x\right), f(x)\right)$, while theory development in previous sections bases on the loss $\ell$ which has the specific form

$$\ell\left(\hat{f}\left(x\right), f\left(x\right)\right) = \sum_{i=1}^{C} l\left(\hat{f}\left(x\right), i\right) f\left(x, i\right). \tag{10}$$

To make it more consistent, we discuss under which condition the Hellinger loss is in the family defined in (10). It is evident that if the labeling function $f$ satisfying $f\left(x, i\right) > 0, \forall x \sim \mathbb{P}$ and $i \in [C]$, for example, we apply label smoothing [10, 9] on ground-truth labels, the Hellinger loss is in the family of interest. That is because the following derivation:

$$D_{1/2}\left(\hat{f}\left(x\right), f(x)\right) = 2\sum_{i=1}^{C}\left[\sqrt{\hat{f}\left(x, i\right)} - \sqrt{f\left(x, i\right)}\right]^2$$

$$= 2\sum_{i=1}^{C}\left[\sqrt{\frac{\hat{f}\left(x, i\right)}{f\left(x, i\right)}} - 1\right]^2 f\left(x, i\right),$$

where we consider $l\left(\hat{f}\left(x\right), i\right) = \left[\sqrt{\frac{\hat{f}(x,i)}{f(x,i)}} - 1\right]^2$.

# 4 Appendix D: Additional Experiments

## 4.1 Experiment on Colored MNIST

### 4.1.1 Dataset

We conduct experiments on the colored MNIST dataset [1] whose data is generated as follow. Firstly, for any original image $X$ in the MNIST dataset [7], the value of digit feature is $Z_d = 0$ if the image's digit is from $0 \rightarrow 4$, while $Z_d = 1$ is assigned to image with digit from $5 \rightarrow 9$. Next, the ground-truth label for the image $X$ is also binary and sampled from either $\mathbb{P}(Y|Z_d = 1)$ or $\mathbb{P}(Y|Z_d = 0)$, depending on the value of digit feature $Z_d$. These binomial distributions are such that $\mathbb{P}(Y = 1|Z_d = 1) = \mathbb{P}(Y = 0|Z_d = 0) = 0.75$. Next, the color feature binary random variable $Z_c$ is assigned to each image conditioning on its label, i.e., $z_C \sim \mathbb{P}(Z_C|Y = 1)$ or $z_c \sim \mathbb{P}(Z_C|Y = 0)$ with $\mathbb{P}(Z_c = 1|Y = 1) = \mathbb{P}(Z_c = 0|Y = 0) = \theta$, depending on the domain. Finally, we color the image red if $Z_c = 0$ or green if $Z_c = 1$.

For both DG and MSDA experiments, there are 7 source domains generated by setting $\mathbb{P}(Z_c = 1|Y = 1) = \mathbb{P}(Z_c = 0|Y = 0) = \theta^{S,i}$ where $\theta^{s,i} \sim Uni\left([0.6, 1]\right)$ for $i = 1, \ldots, 7$. In our actual implemetation, we take $\theta^{s,i} = 0.6 + \frac{0.4}{7}\left(i - 1\right)$. The two target domains are created with $\theta^{T,i} \in \{0.05, 0.7\}$ for $i = 1, 2$. After domain creation, data from each domain is split into training

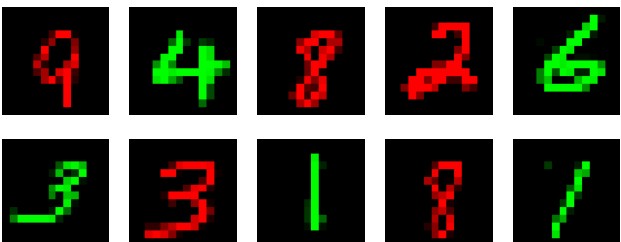

Figure 1: Images in Colored MNIST dataset are "colored" according to color feature $Z_C$. If $Z_C = 0$, channel 0 is kept while channel 1 contains all 0, corresponding to red images. Similarly, $Z_C = 1$ means channel 1 is kept intact while channel 0 is zero-out, represented by green images.

set and validation set. For DG experiment, no data from the target domain is used in training. On the other hand, the same train-validation split is applied to target domains in MSDA and the unlabeled training splits are used for training, while the validation splits are used for testing.

### 4.1.2 Model

We train a hypothesis $\hat{f} = \hat{h} \circ g$ and minimize the classification loss w.r.t. entire source data

$$\mathcal{L}_{gen} := \sum_{i=1}^{7} \frac{N_i}{N_S} \mathbb{E}_{(x,y) \sim \mathbb{D}^{S,i}} \left[ CE \left( \hat{h} \left( g \left( x \right) \right), y \right) \right],$$

where CE is the cross-entropy loss, $N_i$ is the number of samples from domain $i$, and $N_S$ is the total number of source samples.

To align source-source representation distribution, we apply adversarial learning [4] as in [3], in which a min-max game is played, where the domain discriminator $\hat{h}^{s-s}$ tries to predict domain labels from input representations, while the feature extractor (generator) $g$ tries to fool the domain discriminator, i.e., $\min_g \max_{\hat{h}^{s-s}} \mathcal{L}_{disc}^{s-s}$. The source-source compression loss is defined as

$$\mathcal{L}_{disc}^{s-s} := \sum_{i=1}^{7} \frac{N_i}{N_S} \mathbb{E}_{x \sim \mathbb{P}^{S,i}} \left[ -CE \left( \log \hat{h}^{s-s} \left( g \left( x \right) \right), i \right) \right],$$

where $i$ is the domain label. It is well-known [4] that if we search $\hat{h}^{s-s}$ in a family with infinite capacity then

$$\max_{\hat{h}^{s-s}} \mathcal{L}_{disc}^{s-s} = JS \left( \mathbb{P}_g^{S,1}, ..., \mathbb{P}_g^{S,7} \right).$$

.

Similarly, alignment between source and target feature distribution is enforced by employing adversarial learning between another discriminator $\hat{h}^{s-t}$ and the encoder $g$, with the objective is $\min_g \max_{\hat{h}^{s-s}} \mathcal{L}_{disc}^{s-t}$. The loss function for source-target compression is

$$\mathcal{L}_{disc}^{s-t} := \frac{N_k}{N_S + N_T} \mathbb{E}_{x \sim \mathbb{P}^{\pi,S}} \left[ -\log \hat{h}^{s-t} \left( g \left( x \right) \right) \right] + \frac{N_T}{N_S + N_T} \mathbb{E}_{x \sim \mathbb{P}^T} \left[ -\log \left( 1 - \hat{h}^{s-t} \left( g \left( x \right) \right) \right) \right],$$

where $\mathbb{P}^{\pi,S} = \sum_{i=1}^{7} \frac{N_i}{N_S} \mathbb{P}^{S,i}$ is the source mixture and $\mathbb{P}^T$ is the chosen target domain among the two.

Finally, for DG we optimize the objective

$$\min_g \left( \min_{\hat{h}} \mathcal{L}_{gen} + \lambda \max_{\hat{h}^{s-s}} \mathcal{L}_{disc}^{s-s} \right), \tag{11}$$

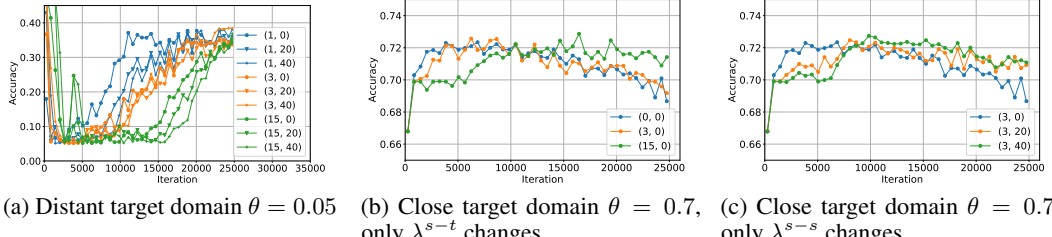

(a) Distant target domain $\theta = 0.05$    (b) Close target domain $\theta = 0.7$, only $\lambda^{s-t}$ changes    (c) Close target domain $\theta = 0.7$, only $\lambda^{s-s}$ changes

Figure 2: Accuracy on distant and close target domains, where tuples $(\lambda^{s-t}, \lambda^{s-s})$ indicate strength of source-target compression and source-source compression, respectively.

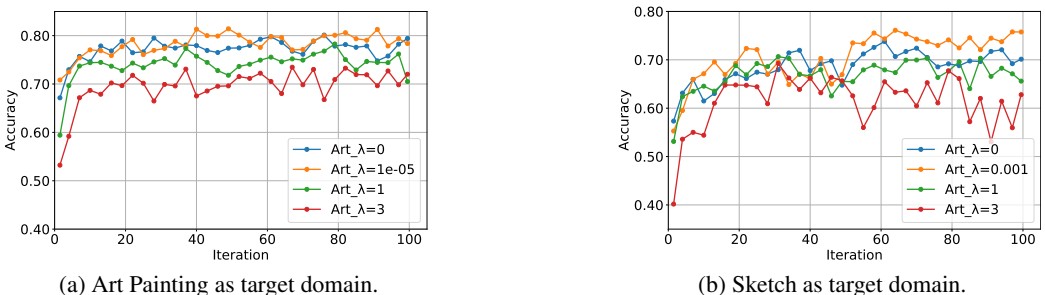

(a) Art Painting as target domain.        (b) Sketch as target domain.

Figure 3: Domain generalization on PACS with different compression strength. On both domain, a slight compression is beneficial which increases accuracy, in-line with possible benefit of compressed DI representation. However, larger compression deteriorates performance, which confirm our discussion using trade-off bound.

where $\lambda$ being the trade-off hyperparameter: $\lambda = 0$ corresponds to DG's general DI representation, while larger $\lambda$ corresponds to more compressed DI representation. On the other hand, the objective for MSDA setting is

$$\min_{g} \left( \min_{\hat{h}} \mathcal{L}_{gen} + \lambda^{s-s} \max_{\hat{h}^{s-s}} \mathcal{L}_{com}^{s-s} + \lambda^{s-t} \max_{\hat{h}^{s-t}} \mathcal{L}_{com}^{s-t} \right), \qquad (12)$$

with $\lambda^{s-s}$ controls the source-source compression and $\lambda^{s-t}$ controls the source-target compression.

Our implementation is based largely on Domain Bed repository [5]. Specifically, the encoder $g$ is a convolutional neural network with 4 cnn layers, each is accompanied by RELU activation and batchnorm, while the classifier $\hat{h}$ and discriminators $\hat{h}^{s-s}, \hat{h}^{s-t}$ are densely connected multi-layer perceptions with 3 layers. Our code can be found in the zip file accompanying this appendix. Moreover, our experiments were run on one Tesla V100 GPU and it took around 30 minutes for one training on Colored MNIST.

### 4.1.3 Multiple-source Domain Adaptation

In additional to DG experiment provided in Section 3 in the main paper, further experiment is conducted on MSDA, in which source-target compression is applied in additional to source-source compression. Specifically, unlabeled data from a target domain is supplied for training, whose label is 1 while all labeled source data has label 0, and the source-target discriminator is tasked with classifying them. We experiment with 2 target domains $\theta^{T,i} \in \{0.5, 0.7\}$ separately and report accuracy on target domain for different compression strength. The result is presented in Figure 2.

It is evident from the figure that the more compression on both source-source and source-target representation, the lower the accuracy. This result aligns with our previous bound (Theorem 8 in main paper), i.e.,

$$d_{1/2}\left(\mathbb{P}^\pi_{\mathcal{Y}}, \mathbb{P}^T_{\mathcal{Y}}\right) - \sum_{i=1}^{K}\sum_{j=1}^{K} \frac{\sqrt{\pi_j}}{K} d_{1/2}\left(\mathbb{P}^T_g, \mathbb{P}^{S,i}_g\right) - \sum_{i=1}^{K}\sum_{j=1}^{K} \frac{\sqrt{\pi_j}}{K} d_{1/2}\left(\mathbb{P}^{S,i}_g, \mathbb{P}^{S,j}_g\right)$$

$$\leq \left[\sum_{i=1}^{K} \pi_i \mathcal{L}\left(\hat{h}\circ g, f^{S,i}, \mathbb{P}^{S,i}\right)\right]^{1/2} + \mathcal{L}\left(\hat{h}\circ g, f^T, \mathbb{P}^T\right)^{1/2}.$$

When source-source and source-target compression is applied, the term $\sum_{i=1}^{K}\sum_{j=1}^{K} \frac{\sqrt{\pi_j}}{K} d_{1/2}\left(\mathbb{P}^T_g, \mathbb{P}^{S,i}_g\right) + \sum_{i=1}^{K}\sum_{j=1}^{K} \frac{\sqrt{\pi_j}}{K} d_{1/2}\left(\mathbb{P}^{S,i}_g, \mathbb{P}^{S,j}_g\right)$ is minimized, raising the lower bound of the loss terms. Subsequently, as source loss $\sum_{i=1}^{K} \pi_i \mathcal{L}\left(\hat{h}\circ g, f^{S,i}, \mathbb{P}^{S,i}\right)$ is minimized, the target loss $\mathcal{L}\left(\hat{h}\circ g, f^T, \mathbb{P}^T\right)$ is high and hence performance is hindered.

However, the drop in accuracy for large compression on close target domain is not as significant as in distant target domain, as indicated in Figure 2b and 2c. In fact, accuracy for some compressed representation is higher than no compressed representation at larget iteration. It is possible that the negative effect of raising lower bound as in trade-off Theorem is counteracted by the benefit of compressed DI representation (Section 2.3.3 of main paper), i.e., the learned classifier for compressed DI representation better approximates the ground-truth labeling function. On the other hand, model with general DI representation cannot approximate this ground-truth labeling function as accurately, but overfit to training dataset, resulting in target accuracy drop at larger iteration.

## 4.2 Experiment on PACS dataset

In order to verify our theoretical finding on real dataset, we conduct further experiment on PACS dataset [8], which has 4 domains: Photo, Art Painting, Cartoon, and Sketch. Among the 4 domains, Photo and Cartoon are chosen as training domains, while Art Painting is chosen as target domain close to the training ones, and Sketch is the target domain distant from the training ones. We use Resnet18 [6] as the feature map, while label classifier and domain discriminator are multi-layer perceptrons. We only investigate DG setting on this dataset, in which training objective function is Eq. 11. The result in Figure 3 illustrates similar pattern to DG experiment on Colored MNIST. Specifically, the accuracies for both target domains raise until a peak is reached and then decrease, which confirms our developed theory for benefit and trade-off of compressed DI representation.