# OpenReview forum: "On Learning Domain-Invariant Representations for Transfer Learning with Multiple Sources"
_NeurIPS.cc/2021/Conference — NeurIPS 2021 Poster_

### Official Review · Reviewer_dgwj · 2021-07-03

**Rating:** 7
**Confidence:** 4

**Summary:**

This work proposes a principled theoretical framework for Multi-Source Domain Adaptation (MSDA) and Domain Generalization (DG), which is lacked in existing works, particularly under a generally setting of multi-class classification. Specifically, this works propose two target domain generalization bounds corresponding to general domain-invariant (DI) representations and compressed DI representations. On such basis, detailed theoretical analysis are proposed to analyze the characteristics of these two bounds in practice and the trade-off between them. By instantiating these two learning objectives under an adversarial domain adaptation framework, authors conduct experiments on colored MNIST and PACS datasets to verify the existence of trade-off and the various generalization capability on different target domains.

**Limitations And Societal Impact:**

The limitations of the proposed method are less discussed in the current version. Also, the potential social impact is lacked in the paper. Though it is a theoretical work, authors are also encouraged to analyze it can potentially benefit and also hamper the related real-world applications when the theory is implemented in a practical system.

**Main Review:**

Strength:

1. This work fills the blank of a general theoretical framework for MSDA and DG problems under a multi-class classification setting.
2. Various previous practical works can be interpreted under this framework as learning general DI representations and compressed DI representations. More future explorations can be inspired by these developed theorems.
3. Empirical results on two datasets verify the theoretical results under both MSDA and DG settings.


Weakness and question:

1. I'm basically convinced by the two proposed target domain generalization bounds. However, in order to better judge the contribution of these bounds, I think it will be good to more completely discuss the connection and difference with the bounds developed in previous theoretical works for MSDA [a,b].
2. It is good to see that the instantiation under the adversarial domain adaptation framework (equivalent to JS divergence) matches the proposed theorems about trade-off and target domain generalization ability. However, I think it would be better to implement one or two more instantiations, e.g. using MMD or WS distance, which makes the empirical verification more convincing.
3. Some typos in the current version:
     - In Line 88 of main paper, [\pi_i]_{i=1}^{C}  ->  [\pi_i]_{i=1}^{K}.
     - In Line 274 of appendix, \max_{\hat{h}^{s-s}}  ->  \max_{\hat{h}^{s-t}}.

[a] Yishay Mansour, Mehryar Mohri, and Afshin Rostamizadeh. 'Domain adaptation with multiple sources.' NeurIPS, 2009.

[b] Judy Hoffman, Mehryar Mohri, and Ningshan Zhang. 'Algorithms and theory for multiple- source adaptation.' NeurIPS, 2018.


Justification of rating:

I am convinced by the proposed theoretical framework and think it could be a solid foundation for the future practical works. I admire the great efforts of authors on boosting the theoretical understanding of MSDA and DG.


**Post Rebuttal**

I have read the response from authors and other reviews. In the response, authors thoroughly discuss the connections and differences with two closely related works, which makes me more convinced about the theoretical contributions of this work. I'll keep my positive rating, and I suggest the authors to supplement additional experimental verifications in the revised version to make these theories more grounded.

**Time Spent Reviewing:**

3

---

> ### Author Response · Authors · 2021-08-10
> **Added comparison to Mansour  et al. (2009) and Hoffman et al. (2018)**
>
> Thank you so much for your review and the constructive suggestion on a more in-depth comparison to previous works. We would like to discuss the two mentioned works and our work here to better illustrate the novelty of our paper.
>
> ## Comparison to Mansour et al. 2009, Hoffman et al. 2018
>
> **Both Mansour et al. (2009) and Hoffman et al. (2018) aim to answer the question
> of combining good source experts into a hypothesis that might perform
> well on target domain with a small label shift compared to the source
> domains**. In particular, Mansour et al. (2009) assumes exactly
> the same labeling function for all souce domains, and target domain
> is an unknown mixture of the source domains. Then, given a set of
> source expert hypotheses (with loss is at most $\epsilon$ on respective
> domain), they show that there exists a distributional weighted combination
> of these experts performing well on target domain with loss at
> most $\epsilon$. Hoffman et al. (2018) further extends this idea to
> a more general case: the source domains could have different labeling
> functions, and the target domain is arbitrary. They show that under
> this setting, there exists an (extended) distributional weighted combination
> of source experts such that the loss on target domain is bounded by
> a source loss term and a discrepancy term between target domain
> and source mixture. In addition, the discrepancy term is measured by Renyi
> divergence on the joint space $\mathcal{X}\times\mathcal{Y}$, and
> it works under more general scenarios with stochastic labeling functions.
> Since the results of Hoffman et al. (2018) can recover the main results
> of Mansour et al. (2009), we focus on compare our work with the results in Hoffman et al. (2018).
>
> First of all, the research questions addressed in their works and
> ours are completely different, leading to different results. Specifically,
> while their works concern performance of combinations of source experts,
> **we focus on investigate the representation learning side in the context
> of multi-source transfer learning**. This is reflected in the fact that
> their bounds only need a single discrepancy term on the joint space
> to quantify domain difference, while our bounds explicitly breakdown
> the domain difference into label shift term and data shift term. Furthermore,
> we measure data shift on feature space, while our label shift and
> loss term are on data space. This allows us to connect the common
> practices of feature alignment in deep learning to the true loss on
> both target and source domains. To the best of our knowledge, our
> upper bounds are the first one with this property, since most previous
> works on DA and MSDA (including these 2 mentioned works) consider both
> loss term and data shift term on data space. Hence, their bounds do
> not directly motivate feature alignment. If one attempts to explain
> the feature alignment by applying these bounds onto feature space,
> one must inevitably obtain the loss term on feature space, which is
> not the learning goal. Moreover, the encoder will be completely left
> out of the equation. Finally, our bounds motivate us to investigate
> further the characteristics of representation space under (a) minimization
> of source loss term, or (b) both source loss term and source feature
> alignment term in Section 2.3 in our paper.
>
> Secondly, while Hoffman et al. (2018) bounds the target loss for any combination
> of source experts, we bound the target loss for any hypothesis, and
> hence our hypothesis set is much more general. Regarding the setting,
> Mansour et al. (2009) is confined to binary classification/regression,
> deterministic labeling functions, L1 loss function, mixture target domain
> and no label-shift. On the other hand, both the work of Hoffman et
> al. (2018) and ours consider stochastic labeling functions, bounded
> convex loss, arbitrary target domain with both label-shift and data-shift.
> Moreover, our bounds hold for multi-label classification problem, while
> Hoffman et al. (2018) considers only binary classification/regression.
>
> ## On experimental evaluation with more distances
>
> Thanks for your suggestion which definitely helps to strengthen our work. We totally agree with your comment that more diverse instantiations would make
> the empirical verification more grounded. However, due to the time
> constraint, we are unable to finalize this experiment. We hope to include
> these further results in an extended version of our paper.
>
> ## On typos and clarification
>
> Thank you so much, we have updated our paper according to your suggestion.

---

### Official Review · Reviewer_kHSY · 2021-07-15

**Rating:** 6
**Confidence:** 4

**Summary:**

This paper studies theoritical aspects of two practical domain adaptation settings, i.e. multiple source domain adaptation and domain generalization settings, where there are multiple source domains and the target domain is potential unavailable during training.
Specifically, novel upper-bounds for the target general loss are developed and proved. Based on the upper bounds, two kinds of domain-invariant representations are defined. The pros and cons are also studied and the trade-offs of enforcing learning each domain-invariant representation are investigated.
Experiments are conducted to inspect the trade-off of these representations for offering practical hints regarding how to use them in practice and explore other interesting properties of the developed theory.

**Limitations And Societal Impact:**

There is insufficient discussion on the limitations and potential negative societal impact of the work.
1. There already exist some theoritical analysis, such as [18] [50] on multi-source domain adaptation. However, there is no clear comparison between the upper bounds of the proposed method and these existing methods. Are the designed upper bounds tighter than exsisting ones?
2. The experimental evaluation is weak. Generally, there are many datasets to evaluation the performances of multiple source domain adaptation and domain generalization algorithms, such as Digits-five, Office-31, Office-Home, DomainNet. However, there is no experiment on these datasets and no comparison with existing methods, such as [50] [52].
3. There is no analysis on the conditions on which the proposed upper bounds hold. If the conditions are not satisfied, what will happen?
4. Comparison with exisiting domain adaptation in related work section is not satisfying. Adding some recent surveys would make more sense, such as "A review of domain adaptation without target labels", "A Review of Single-Source Deep Unsupervised Visual Domain Adaptation", "A Survey of Unsupervised Deep Domain Adaptation",  "Multi-source Domain Adaptation in the Deep Learning Era: A Systematic Survey".

**Main Review:**

Originality: This paper focuses on the theory analysis of multiple source domain adaptation and domain generalization. The developed upper bounds are novel and make sense. The defined two kinds of domain-invariant representations are reasonable. The pros and cons and the trade-offs of enforcing learning each domain-invariant representation are also well explored. The theory part is sufficient.

Quality: The developed upper bounds sound good and the corresponding technical domain-invariant representation is technically sound. The main claims are supported mainly by theoritical results. The authors prove and explain the theoritical strengths well but the discussion on the weaknesses of the propoded method and the experiment are insufficient.

Clarity: Generally, this paper is well organized and clearly written. But there are too many mathetical notations and equations, which make it difficult to understand for general domain adaptation researchers.

Significance: The novel upper bounds for multiple source domain adaptation and domain generalization are important and can motivate other researchers or practitioners likely to use the upper bound to design novel algorithms.


**Time Spent Reviewing:**

4 hours

---

> ### Author Response · Authors · 2021-08-10
> **Comparison to Zhao et al. 2018, Hoffman et al. 2018. Discussion on conditions of the bounds. Added experiment on VLCS.**
>
> We thank the reviewer for your thoughtful review!! We hope to address some of your comments as presented below.
>
> ## Comparison to Hoffman et al. 2018 and Zhao et al. 2018
>
> ### Motivation of different upper bounds
>
> **Zhao et al. (2018) improves statistical learning bound.**
>
> The upper bounds developed in Zhao et al. (2018)'s paper is a direct extension
> of Ben-David et al. (2010)'s bound to multi-source setting. Specifically,
> their theorem 2 gives the average case classification bound which
> involves (a) the loss on source mixture, (b) the average of discrepancies
> between the target domain and the source domains, (c) the risk of
> the optimal hypothesis on both source mixture and target, and (d)
> the approximation error. This bound is developed in order to improve
> the sampling rate in the approximation error term, which is $\mathcal{\mathcal{O}}\left(\sqrt{1/m}\right)$
> in the naive worst case bound, to $\mathcal{\mathcal{O}}\left(\sqrt{1/km}\right)$
> in the theorem 2's bound.
>
> **Hoffman et al. (2018) focuses on combining source experts**
>
> This work focuses on the question of combining good source experts into a hypothesis
> that might perform well on target domain with small label shift
> compared to the source domains. In particular, they show that there
> exists a distributional weighted combination of source experts such
> that the loss on target domain is bounded by a source loss term and
> a domain discrepancy term between target domain and source mixture.
>
> **Our work focuses on investigating the representation learning side in the context
> of multi-source transfer learning.**
> Specifically, we measure data shift
> on feature space, while both label shift and loss term are on data
> space. This allows us to connect the common practices of feature alignment
> in deep learning to the true loss on both target and source domains.
> To the best of our knowledge, our upper bounds are the first one with
> this property, since most previous works on DA and MSDA (including
> these 2 mentioned works) consider both loss term and data shift term
> on data space. Hence, their bounds do not directly motivate feature
> alignment. If one attempts to explain the feature alignment by applying
> these bounds onto feature space, one must inevitably obtain the loss
> term on feature space, which is not the learning goal. Moreover, the
> encoder will be completely left out of the equation. Finally, our
> bounds motivate us to investigate further the characteristics of representation
> space under (a) minimization of source loss term, or (b) both source
> loss term and source feature alignment term in Section 2.3 in our
> paper.
>
> ### On technical level
>
> In Zhao et al. (2018), the definitions of $\mathcal{H}$-divergence
> and $\mathcal{H}\Delta\mathcal{H}$-divergence rely on the notion
> of characteristic function: $h$ is the characteristic function of
> a set $B\subset\mathcal{X}$ if $h\left(x\right)=1,\forall x\in B$.
> Therefore, these distribution discrepancy can only be defined in binary
> classification/regression problem with deterministic labeling function
> and specific L1 loss function. On the other hand, both Renyi distance
> in Hoffman et al. (2018) and Hellinger distance in our paper are applicable to
> the most general setting: multi-label, stochastic labeling function,
> upper bounded convex loss, as these distances do not require information
> about label set, hypothesis set, and even the loss function.
>
> Secondly, using space of label simplexes, we can offer bounds for
> multi-label classification, while Hoffman et al. (2018)'s bounds work
> for only binary case. We can also separate distribution discrepancy
> into 2 terms: feature discrepancy and label mismatch for better characterize
> learning scenarios, compared to discrepancy on joint space in Hoffman
> et al. (2018). Finally, while they bound the target loss for any
> combination of source experts, we bound the target loss for any hypothesis,
>  hence our hypothesis set is remarkably  more general.
>
> ## On more experimental verification
>
> We definitely agree that more experiments on a diverse range of datasets would
> substantially strengthen the empirical verification. However, since the main contribution of our paper stays in theoretical aspect and we conduct
> experiments to empirically verify theoretical findings,
> we initially experiment on only two datasets in the main paper: Colored MNIST and PACS. Inspired by your suggestion, we run additional experiment on  VLCS dataset using DomainBed code [1]. We couldn't manage to run on different target domain and other datasets due to the time constraint. The result is shown in following table, with mean and standard deviation over 3 runs.
>
> | $\lambda^{s-s}$ | 0 | 1 | 3 | 7 |
> |---|---|---|---|---|
> |Accuracy on Label Me (L) | $0.632 \pm 0.014$ | $0.673 \pm 0.008$ | $0.655 \pm 0.009$ | $0.608 \pm 0.019$|
>
> Here, the setting is domain generalization (DG), source domains are VCS,  target domain is L. The model is DANN with resnet18 backbone. The loss function: $\mathcal{L} =\mathcal{L}^{class} + \lambda^{s-s} \mathcal{L}^{disc}$,
> where $\lambda^{s-s}$ is the source-source compression strength.
>
> Evidently, we observe the same pattern as follows
> * A sufficient compression leads to an increase in target accuracy, due to
> the benefit of compressed domain-invariant feature (Section 2.3.3
> in our paper).
> * Over-compression leads to a decrease in target accuracy, illustrating
> the trade-off bound (Section 2.4 in our paper).
>
> ## The conditions for our upper bounds
>
> We consider a general setting for MSDA and DG problems as illustrated as follows:
>
> 1. The classification problem is multi-label, and we tackle transfer
> learning from multiple source domain to one target domain. Both target
> loss upper bounds in our paper hold for any mixture of any number
> of source domains. Also, there is no restrictrion on the source domains
> and target domain, which means both label-shift and data-shift is
> allowed between domains.
> 2. Secondly, we only assume general upper-bounded loss if the labeling function is deterministic, and upper-bounded, convex loss if the labeling function is stochastic.
> Compared to much more restricted loss functions (such as L1 loss) used in many previous works, ours is much more general, allowing many classification methods to be explained using our theory. Specifically,
> we let $l\left(\hat{f}\left(x\right),y\right)$ be any bounded loss
> between prediction $\hat{f}\left(x\right)$ and the ground-truth label
> $y$. If the ground-truth labeling function is deterministic, i.e., $f:\mathcal{X} \mapsto \mathcal{Y}$, then $y = f(x)$. Meanwhile, $\hat{f}(x)$ is a simplex, e.g., output of the last softmax layer in a neural network. If the ground-truth labeling function is stochastic, i.e., $f:\mathcal{X} \mapsto \mathcal{Y}^{\Delta}$, where $\mathcal{Y}^{\Delta}$ is the space of label simplexes, the ground-truth label is sampled as $y\sim f\left(x\right)$.
> Therefore, we define expected loss on sample $x$ as $\ell\left(\hat{f}\left(x\right), f\left(x\right)\right) = \mathbb{E}_{y\sim f\left(x\right)}\left[l\left(\hat{f}\left(x\right),y\right)\right]$. Since $l$ is bounded, $\ell$ is also bounded, and the expectation
> makes $\ell$ convex on the ground-truth stochastic labeling function $f\left(x\right)$. In the deterministic setting, $\ell\left(\hat{f}\left(x\right),f\left(x\right)\right) = l \left(\hat{f}\left(x\right),f\left(x\right)\right)$
> 3. Moreover, the family of encoder $g$ and family of classifier $h$
> is assumed to be the set of all functions from $\mathcal{X}\mapsto\mathcal{Z}$
> and $\mathcal{Z}\mapsto\mathcal{Y}_{\Delta}$, respectively. In actual
> implementation of encoder and classifier in deep learning, the neural
> networks family could represent a smaller family of functions compared
> to our bounds. Therefore, the minimum upper bound in practice could
> be larger than what is predicted from our bound.
>
>
> ## Related work section
>
> Thank you for this suggestion. In the revised version of our paper,
> we will definitely add suitable survey papers in the Related Work
> section for a more complete comparison.
>
> ### References
> [1] [Ishaan Gulrajani, David Lopez-Paz. In Search of Lost Domain
> Generalization. ICLR 2021](https://openreview.net/forum?id=lQdXeXDoWtI)

---

### Official Review · Reviewer_XYGp · 2021-07-16

**Rating:** 6
**Confidence:** 4

**Summary:**

This paper proposes two novel upper bounds for the target loss in the context of multi-source domain adaptation and generalization and defines two types of domain-invariant representations: general domain invariant and compressed domain invariant.

**Limitations And Societal Impact:**

The authors not addressed these aspects.

**Main Review:**

**Originality**:

Domain adaptation and generalization are very important issues and as a consequence, many theoretical and practical works have been proposed. This submission completes this large avenue of research with a focus on domain invariant representations in the specific setting of multi-source adaptation and generalization.

The submission can be completed with more comparisons and positioning with other theoretical works on multi-source domain adaptation such as the one in ref [50]. For instance,  how is Theorem 1 compared with Theorem 3 of ref [50]?

**Quality**

- All core assumptions are made explicit, their motivation and their implications are discussed. Some insights should be given on the choice of considering multiple sources as a mixture of source distributions. For instance, some recent works have been proposed on domain aggregation in the context of multi-source domain adaptation such as [Wen, ICML 2020](http://proceedings.mlr.press/v119/wen20b.html) and this point could be more discussed. In particular, what are the limitations of considering mixtures of distributions? How it behaves in the context of source disagreement (see for instance [Liu et al, 2021](https://hal.archives-ouvertes.fr/hal-02944558))

- Experiments are proposed in order to support the claims. The experimental setup is straightforward, all relevant details are given. Experiments on the PACS dataset are also given to show how the approach behaves on real data.

- The body of supplementary material is impressive. The proofs are sufficiently easy to follow and well detailed.


**Clarity**:
The paper is well-written and structured. It is a very dense paper. Supplementary materials are mandatory in particular to have the details on the experimental settings.

**Significance**: The submission brings new theoretical results

**Time Spent Reviewing:**

3

---

> ### Author Response · Authors · 2021-08-10
> **Comparison to Zhao et al. (2018) and discussion on mixture choice**
>
> Thank you so much for your thoughtful review and insightful suggestions!! We would like to address some of your concerns as below.
>
> ## Comparison to the work of Zhao et al. 2018 (ref [50])
>
> The work of Zhao et al. (2018) and our work both address the adaptation problem from multiple
> source domains to target domain given labeled source data and unlabeled
> target data. We would like to illustrate the differences between their work and our work at two main points.
>
> ### The research questions are different.
>
> **Zhao et al. (2018) improves statistical learning bound.**
>
> The upper bounds developed in Zhao et al., (2018) is a direct extension
> of Ben-David et al. (2010)'s bounds to multi-source setting. Specifically,
> their theorem 2 gives the average case classification bound which
> involves (a) the loss on source mixture, (b) the average of discrepancies
> between the target domain and the source domains, (c) the risk of
> the optimal hypothesis on both source mixture and target, and (d)
> the approximation error. This bound is developed in order to improve
> the sampling rate in the approximation error term, which is $\mathcal{\mathcal{O}}\left(\sqrt{1/m}\right)$
> in the naive worst case bound, to $\mathcal{\mathcal{O}}\left(\sqrt{1/km}\right)$
> in their theorem 2's bound. Based on this theoretical understanding,
> the authors propose a multi-source extension of DANN method (Ganin et al.
> (2016)), where a domain discriminator is tasked to distinguish
> samples from different domains.
>
> **Our work focuses on explaining representation space.**
>
> * Notably, the upper
> bounds developed in our paper involve (a) the source mixture's loss
> measured on data space, (b) the label-shift term involving labeling
> function from data space to label space, and (c) the domain discrepancy
> on feature space. Hence, our upper bounds directly and explicitly
> motivate the feature alignment in practice in order to lowering true
> target loss as measured on data space. On the contrary, both Ben-David
> et al. (2010)'s upper bound and the MSDA version in Zhao et al. (2018) are designed solely on data space for both the loss term and discrepancy
> term. Hence, they do not directly motivate feature alignment. If one
> attempts to explain the feature alignment by applying these bounds
> onto feature space, one must inevitably obtain the loss term on feature
> space, which is not the learning goal. Moreover, the encoder will
> be completely left out of the equation.
> * Secondly, to further understand the characteristics of different approaches
> in multi-source transfer learning, we define two kinds of representation:
> (a) General DI representation resulted from minimization of source
> mixture loss, and (b) Compressed Di representation resulted from simultaneous
> minimization of source loss and alignment of source-domain feature.
> We then explore how the two different representations shape the feature
> space and what is the benefit in each case. Finally, our lower bound
> shows how compressed representation could hurt target loss when domains'
> label marginals are different.
>
> Due to this difference, we believe  their work and ours are complimentary to each other. Specifically, their improvement on the learning aspect
> might be integrated into our bounds, which involves only population
> losses.
>
> ### The settings are different.
>
> While the bounds in Zhao et al. (2018) are developed for binary classification/regression
> problems with deterministic labeling functions and specific L-1 loss
> functions, our upper bounds work for much more general cases: multi-label
> classification problems with stochastic labeling functions and general
> upper bounded, second-argument-convex loss.
>
> Furthermore, the definitions of $\mathcal{H}$-divergence and $\mathcal{H}\Delta\mathcal{H}$-divergence in their work
> rely on the notion of characteristic function: $h$ is the characteristic
> function of a set $B\subset\mathcal{X}$ if $h\left(x\right)=1,\forall x\in B$.
> Therefore, these distribution discrepancies can only be defined in binary
> classification problems, and are not applicable in multi-label classification
> problems. To tackle this issue, we use Hellinger distance to measure
> distribution discrepancy, which works in the most general classification
> problems as it doesn't require information about label set, hypothesis
> set, and even the loss function.
>
> ## Discussing the choices of mixture
>
> Thank you for suggestion on these interesting works.
>
> Wen et al. (2020) fills in the gap between theoretical analysis
> and practical domain aggregation. They found a trade-off between convergence
> speed (which depends on $m/\|\boldsymbol{\alpha}\|$, where $m$ is
> the sample size and $\boldsymbol{\alpha}$ is mixture weights) and
> the mixture average $\sum_{i}\alpha_{i}\left(\mathcal{L}\left(\hat{f},\mathbb{D}^{S,i}\right)+\text{disc}\left(\mathbb{P}^{T},\mathbb{P}^{S,i}\right)\right)$
> of source losses and target-source distributional discrepancies. On
> the other hand, the loss terms in our bound are population losses
> and hence do not include convergence speed with finite sample size.
> Therefore, our bounds work for all choices of mixture weight $\boldsymbol{\alpha}$.
> Nevertheless, we believe it is a promising further research to develop
> the empirical version of our bounds with added convergence speed.
> In such case, we believe that a similar trade-off as in Wen at el. (2020) could
> be realized.
>
> Concerning the choice of mixture weight with regard to population
> loss, Theorem 3 in our paper suggests placing larger weight $\alpha_{i}$
> on source domain $i$ with smaller source loss $\mathcal{L}\left(\hat{f},\mathbb{D}^{S,i}\right)$
> and smaller feature discrepancy $d_{1/2} \left( \mathbb{P}^T_g, \mathbb{P}^{S,i}_g \right)$,
> concurring in spirit with Wen at al. (2020).
>
> However, the effect of mixture
> weight $\boldsymbol{\alpha}$ in our bound is more complex due to
> the fact that there is an additional feature discrepancy $d_{1/2}\left(\mathbb{P}_g^{S,j},\mathbb{P}_g^{S,i}\right)$
> between the source domains, and the weights of these feature discrepancies
> depend on $\sqrt{\alpha_i}$, making them not conventional weighted
> average.
>
> We might make a guess that the search for optimal choice
> of $\boldsymbol{\alpha}$ that results in smallest upper bound would
> favor similar source domains (smaller $d_{1/2}\left(\mathbb{P}_g^{S,j},\mathbb{P}_g^{S,i}\right)$)
>
> that are all close to target domain (smaller $d_{1/2}\left(\mathbb{P}_g^{T},\mathbb{P}_g^{S,i}\right)$).
> In other word, an aggregation method would choose source domains aligned
> with target on feature space, but avoid disagreement between source
> domains.
>
> Not only that, our lower bound (Theorem 8) seems to suggest choosing source mixture so that label mismatch between target and source-mixture $\mathcal{L}\left(\mathbb{P}^\pi_{\mathcal{Y}}, \mathbb{P}^T_{\mathcal{Y}}\right)$ is small, in order to lower loss on target domain. However, if this choice of mixture weight results in larger target's upper bound, transfer learning to target might not be guaranteed.
> We think that this trade-off between feature mismatch (upper bound) and label mismatch (lower bound) w.r.t. the choice of mixture weight is indeed exciting, and we hope to investigate more on it in the future.
>
> Liu et al. (2021) suggests a combination of source experts based on
> distance between target and source domains on feature space and evidence
> theory. However, the prediction function used in this work is a belief function,
> not conventional probability function, hence we are not sure if the
> target loss could be bounded as in our work. Indeed, to the best of
> our knowledge, there isn't any target bound using evidence theory
> so far, and this extension direction could be an interesting research
> in the future.

---

> ### Comment · Reviewer_XYGp · 2021-08-25
> **Main concerns are adressed**
>
> I have read the rebuttal and the other reviews. I have read the paper again. I appreciate the very precise authors’ responses. I maintain my rating.

---

### Official Review · Reviewer_C95H · 2021-07-22

**Rating:** 5
**Confidence:** 4

**Summary:**

The paper derives theory for invariant representations in unsupervised domain adaptation and domain generalization, in the case when there are multiple labeled source domains.

**Limitations And Societal Impact:**

There has not been any discussion regarding limitations in this paper, and any brief discussion would be welcome.

**Main Review:**

The paper provides a large volume of theory for the multiple-source domain adaptation and domain generalization setting. However, this paper lacks clarity and sufficient comparison with existing work to meet the requirements for acceptance.

First of all, the paper claims to be the first one to theoretically study in detail the invariant representations for domain adaptation in the multiple-source setting. However, the paper by [Zhao et al. 2018] studied exactly this for domain adaptation, and has provided a generalization bound that, from my understanding, is minimized when we acquire a compressed domain-invariant representation (using the authors' terminology). Given the similarity between the proposed study and the one by [Zhao et al. 2018], it is not clear to me what the novelty is in this paper, as the authors have not provided sufficient discussion and comparison.

Furthermore, the authors derive bounds in the case when P(Y) changes across domains. It is obvious that invariant representation learning is not viable for this setting, and it needs to be modified (relaxed) in order to be suitable. This has been studied in [Wu et al. 2019], and it would be great if the authors could discuss the comparison between their reasoning and the one presented in this study.

Finally, the paper can greatly benefit from better clarity and discussion of individual terms introduced in the theoretical results. In particular, Theorem 5 and the paragraph preceding it are confusing since $h$ here represents a domain discriminator, where before and after this results it represents a classifier for the label $y$.

References:
Wu, Yifan, et al. "Domain adaptation with asymmetrically-relaxed distribution alignment." International Conference on Machine Learning. PMLR, 2019.

Zhao, Han, et al. "Adversarial multiple source domain adaptation." Advances in neural information processing systems 31 (2018): 8559-8570.



**Time Spent Reviewing:**

4

---

> ### Author Response · Authors · 2021-08-09
> **Comparison to Zhao et al. 2018 and discussion on label mismatch**
>
> Thank you so much for your thorough review and constructive comments!! We have endeavored to address these comments as much as we can, given the rebuttal guidelines and constraints.
>
> ## Comparison to the work of Zhao et al. 2018
>
> The work of Zhao et al. (2018) and our work both address the adaptation problem from multiple
> source domains to target domain given labeled source data and unlabeled
> target data. However, we believe our development is crucially different
> to theirs, as pointed out as follows.
>
> ### The research questions are different.
>
> **Zhao et al. (2018) improves statistical learning bound.**
>
> The upper bounds developed in Zhao et al., (2018) is a direct extension
> of Ben-David et al. (2010)'s bounds to multi-source setting. Specifically,
> their theorem 2 gives the average case classification bound which
> involves (a) the loss on source mixture, (b) the average of discrepancies
> between the target domain and the source domains, (c) the risk of
> the optimal hypothesis on both source mixture and target, and (d)
> the approximation error. This bound is developed in order to improve
> the sampling rate in the approximation error term, which is $\mathcal{\mathcal{O}}\left(\sqrt{1/m}\right)$
> in the naive worst case bound, to $\mathcal{\mathcal{O}}\left(\sqrt{1/km}\right)$
> in their theorem 2's bound. Based on this theoretical understanding,
> the authors propose a multi-source extension of DANN method (Ganin et al.
> (2016)), where a domain discriminator is tasked to distinguish
> samples from different domains.
>
> **Our work focuses on explaining representation space.**
>
> * Notably, the upper
> bounds developed in our paper involve (a) the source mixture's loss
> measured on data space, (b) the label-shift term involving labeling
> function from data space to label space, and (c) the domain discrepancy
> on feature space. Hence, our upper bounds directly and explicitly
> motivate the feature alignment in practice in order to lowering true
> target loss as measured on data space. On the contrary, both Ben-David
> et al. (2010)'s upper bound and the MSDA version in Zhao et al. (2018) are designed solely on data space for both the loss term and discrepancy
> term. Hence, they do not directly motivate feature alignment. If one
> attempts to explain the feature alignment by applying these bounds
> onto feature space, one must inevitably obtain the loss term on feature
> space, which is not the learning goal. Moreover, the encoder will
> be completely left out of the equation.
> * Secondly, to further understand the characteristics of different approaches
> in multi-source transfer learning, we define two kinds of representation:
> (a) General DI representation resulted from minimization of source
> mixture loss, and (b) Compressed Di representation resulted from simultaneous
> minimization of source loss and alignment of source-domain feature.
> We then explore how the two different representations shape the feature
> space and what is the benefit in each case. Finally, our lower bound
> shows how compressed representation could hurt target loss when domains'
> label marginals are different.
>
> Due to this difference, we believe  their work and ours are complimentary to each other. Specifically, their improvement on the learning aspect
> might be integrated into our bounds, which involves only population
> losses.
>
> ### The settings are different.
>
> While the bounds in Zhao et al. (2018) are developed for binary classification/regression
> problems with deterministic labeling functions and specific L-1 loss
> functions, our upper bounds work for much more general cases: multi-label
> classification problems with stochastic labeling functions and general
> upper bounded, second-argument-convex loss.
>
> Furthermore, the definitions of $\mathcal{H}$-divergence and $\mathcal{H}\Delta\mathcal{H}$-divergence in their work
> rely on the notion of characteristic function: $h$ is the characteristic
> function of a set $B\subset\mathcal{X}$ if $h\left(x\right)=1,\forall x\in B$.
> Therefore, these distribution discrepancies can only be defined in binary
> classification problems, and are not applicable in multi-label classification
> problems. To tackle this issue, we use Hellinger distance to measure
> distribution discrepancy, which works in the most general classification
> problems as it doesn't require information about label set, hypothesis
> set, and even the loss function.
>
> ## Discussing label mismatch cases and work of Wu et al. (2019)
>
> Thank you for the suggestion of this insightful paper. It addresses
> in-depth the label marginal mismatch scenario in (single-source) DA
> setting, under binary classification problem with domain invariant,
> deterministic ground-truth labeling function. The paper first shows
> that if there is a difference in label marginals, enforcing hard feature
> alignment on feature space, i.e., $\mathbb{P}^S_{\phi} = \mathbb{P}^T_{\phi}$ ,
> where $\phi$ is the encoder, will lead to an inevitable mismatch
> between source's and target's feature space labeling functions. Hence,
> the loss on target domain will be high. To alleviate the problem,
> the authors suggest a method of relaxed feature alignment in which
> support of source domain contains the support of target domain on
> feature space. However, they note that learning such a representation
> space does not guarantee low target loss, and the theoretical guarantee
> of aligned feature distribution is still an open question (the quote by
> the authors). Nevertheless, under a specific assumption of connectedness
> from target to source domain on data space, the authors show that
> a Lipschitz encoder can always be constructed, which leads to such
> relaxed feature alignment. Then, in such case, an upper bound on target
> loss is provided, which is controlled by the target-source connectedness
> on data space and the separation between support sets of class-conditional
> feature distributions of source domains.
>
> While Wu et al. (2019)'s research focuses on single-source, binary classification
> DA under specific constraints, we seek out to address a much broader
> setting in multi-source transfer learning problem. We also investigate
> the effect of feature alignment, both source-source alignment and
> source-target alignment via our bounds. Our results show that target
> loss on data space is indeed upper bounded by a discrepancy on feature
> space, which is novel compared to other previous works in MSDA. Moreover,
> target loss is also lower bounded by a combination of label mismatch $\mathcal{L}\left(\mathbb{P}^\pi_{\mathcal{Y}}, \mathbb{P}^T_{\mathcal{Y}}\right)$,
> feature discrepancy, and source loss. This latter result agrees in
> the spirit with Wu et al. (2019)'s discussion.
> Interestingly, our lower bound seems to suggest choosing source mixture so that label mismatch between target and source-mixture $\mathcal{L}\left(\mathbb{P}^\pi_{\mathcal{Y}}, \mathbb{P}^T_{\mathcal{Y}}\right)$ is small, in order to lower loss on target domain. However, if this choice of mixture weight results in larger target's upper bound, transfer learning to target might not be guaranteed.
> We think that this trade-off between upper bound and lower bound w.r.t. the choice of mixture weight is indeed exciting, and we hope to investigate more on it in the future.
>
> Extension of Wu et al. (2019)'s idea to our setting
>
> However, due to the already dense paper, we have not addressed and suggested
> a method to overcome label mismatch in our setting of multi-label,
> multi-source, stochastic labeling function, with non-zero label shift.
> We believe that this question is non-trivial to address, and could
> be a solid theoretical research work in the future. Specifically,
> while it is possible to reason about effect of feature alignment when
> there is a label mismatch in binary, deterministic labeling function,
> zero label-shift (Proposition 3.1 and 3.2 in Wu et al. 2019), the
> combination of multi-label, stochastic labeling, and non-zero label-shift
> assumptions might create intractable dynamics affecting target loss.
> Hence, representation learning in such general case becomes very complex,
> and we may not currently have a reasonable suggestion.
>
> ## Better clarity and discussion
>
> Thank you for the suggestion, we will try our best to clarify and
> polish our paper for improving its comprehensibility.

---

> ### Author Response · Authors · 2021-08-29
> **We wonder if we have partly addressed your concerns**
>
> Dear Reviewer C95H,
>
> Thank you for your time helping to review and improve the paper.
>
> As the discussion period is nearing an end, thus we wonder if you can spend some time going over our newest replies, to see if they successfully answered your questions or not. This is also to give us a decent amount of time to address any of your remaining/additional concerns.
>
> Best regards,
>
> Authors.

---

### Decision · Program_Chairs · 2021-09-28

**Decision:**

Accept (Poster)

**Comment:**

This paper develops novel upper bounds for the target risk in the multiple source domain adaptation/generalization settings. Based on the theoretical results, the authors further study how to learn proper domain invariant representations. The proposed theory provides new insights and offers practical hints on multiple source domain adaptation. I thus recommend acceptance of this paper.

Nevertheless, there are some concerns that need to be addressed in the final version. The main concern is the comparison with existing theoretical results and methods in Zhao et al. 2018 and Hoffman et al. 2018. The authors have well addressed these in the rebuttal, and I hope the authors could carefully revise the paper to incorporate these discussions.


**Consistency Experiment:**

NeurIPS has a long history of experimentation. In 2014, NeurIPS ran an experiment in which 10% of submissions were reviewed by two independent committees to quantify the randomness in the review process. This year, we repeated a variant of this experiment to see how the quality of the review process has changed over time.  This paper was part of the experiment and was therefore assigned to two committees (consisting of reviewers, an Area Chair, and a Senior Area Chair) that reached independent decisions.  If both committees made the same recommendation, this recommendation was followed. If a single committee recommended acceptance, the paper was accepted (with the exception of a few cases in which the other committee identified what we considered a fatal flaw, e.g., an error in a key result).

This copy’s committee reached the following decision: **Accept (Poster)**

The other committee assigned to the paper recommended **Reject**.  You can find the other set of reviews, along with any follow up discussion with the authors here:
https://openreview.net/forum?id=gH1ATCT7RiK